DOI: 10.1038/s41467-018-06897-5　　**OPEN**

# Spatiotemporal regulation of the GPCR activity of BAI3 by C1qL4 and Stabilin-2 controls myoblast fusion

Noumeira Hamoud[1,2], Viviane Tran[1,3], Takahiro Aimi[4,5], Wataru Kakegawa[4,5], Sylvie Lahaie[1,6], Marie-Pier Thibault[1], Ariane Pelletier[1], G. William Wong[7,8], In-San Kim[9,10], Artur Kania [1,6,11], Michisuke Yuzaki [4,5], Michel Bouvier [3,12] & Jean-François Côté [1,2,3,11]

Myoblast fusion is tightly regulated during development and regeneration of muscle fibers. BAI3 is a receptor that orchestrates myoblast fusion via Elmo/Dock1 signaling, but the mechanisms regulating its activity remain elusive. Here we report that mice lacking BAI3 display small muscle fibers and inefficient muscle regeneration after cardiotoxin-induced injury. We describe two proteins that repress or activate BAI3 in muscle progenitors. We find that the secreted C1q-like1–4 proteins repress fusion by specifically interacting with BAI3. Using a proteomic approach, we identify Stabilin-2 as a protein that interacts with BAI3 and stimulates its fusion promoting activity. We demonstrate that Stabilin-2 activates the GPCR activity of BAI3. The resulting activated heterotrimeric G-proteins contribute to the initial recruitment of Elmo proteins to the membrane, which are then stabilized on BAI3 through a direct interaction. Collectively, our results demonstrate that the activity of BAI3 is spatio-temporally regulated by C1qL4 and Stabilin-2 during myoblast fusion.

[1] Institut de Recherches Cliniques de Montréal (IRCM), Montréal, QC H2W 1R7, Canada. [2] Département de Médecine (Programmes de Biologie Moléculaire), Université de Montréal, Montréal, QC H3T 1J4, Canada. [3] Département de Biochimie, Université de Montréal, Montréal, QC H3T 1J4, Canada. [4] Department of Physiology, Keio University School of Medicine, Tokyo 160-8582, Japan. [5] Core Research for Evolutional Science and Technology (CREST), Japan Science and Technology Agency (JT), Tokyo 102-0075, Japan. [6] Integrated Program in Neuroscience, McGill University, Montréal, QC H3A 2B4, Canada. [7] Department of Physiology, Johns Hopkins University School of Medicine, Baltimore, MD 21205, USA. [8] Center for Metabolism and Obesity Research, Johns Hopkins University School of Medicine, Baltimore, MD 21205, USA. [9] Biomedical Research Institute, Korea Institute Science and Technology, Seoul 136-791, Republic of Korea. [10] KU-KIST school, Korea University, Seoul 136-701, Republic of Korea. [11] Department of Anatomy and Cell Biology, McGill University, Montréal, QC H3A 1A3, Canada. [12] Institut de Recherches en Immunologie et Cancérologie (IRIC), Université de Montréal, Montréal, QC, Canada H3C 3J7. Correspondence and requests for materials should be addressed to J.-F.C. (email: jean-francois.cote@ircm.qc.ca)

Fusion of myoblasts during embryonic myogenesis, or of satellite cell-derived myoblasts during muscle regeneration, is central to the formation of multinucleated fibers[1–3]. The molecular mechanisms controlling myoblast fusion remains poorly defined. By merging the power of genetics and tissue imaging, *Drosophila* studies led the way in the identification of genes controlling myoblast fusion during embryogenesis. The current view is that cell adhesion receptors activate signaling pathways that engage actin, allowing myoblast fusion[4]. While less is known about myoblast fusion in vertebrates, orthologues of the fly proteins, including guanine nucleotide exchange factor Dock1, GTPase Rac1, and actin nucleator N-Wasp, have an evolutionarily conserved essential role in fusion in mice[5–7]. Proteins involved in cell–cell or cell–matrix adhesion, including Cdon, M/N-Cadherins, Neogenin, and Integrin ß1, also contribute to the myoblast differentiation and fusion[8–11]. How these factors work together to promote fusion remains to be defined.

Recently, vertebrate membrane associated proteins orchestrating fusion have been uncovered. Myomaker, a myoblast specific protein with fusogenic activity, was found to be vital for fusion[12,13]. *MYOMAKER* mutations are responsible for the Carey–Fineman–Ziter syndrome, a group of congenital myopathies that originate from defective myoblast fusion[14]. The microprotein Myomixer (Myomerger/Minion) is also expressed at the time of fusion and is essential for myoblast fusion in vivo[15–17]. Stabilin-2 was identified as a phosphatidylserine receptor expressed during myoblast differentiation[18] that transduces the pro-fusion signals triggered by non-apoptotic phosphatidylserine exposed by myoblasts[19]. The G-protein Coupled Receptors (GPCRs) BAI1 and BAI3 were found to promote myoblast fusion by interacting with the Elmo/Dock complex[20,21]. Notably, the molecular mechanisms that ensure the regulation of the pro-fusion activity of BAI proteins are unknown.

BAI1–3 belong to the family of Adhesion GPCRs that are defined by long extracellular and intracellular domains[22]. They contain thrombospondin repeats (TSRs) in their extracellular domains as well as an Elmo-binding site (EBS) in their intracellular tail[20,23]. The presence of a GPCR Auto-proteolysis-Inducing (GAIN) domain is a signature of Adhesion GPCRs[22,24,25]. Autocleavage of Adhesion GPCRs contributes to their ability to activate heterotrimeric G-proteins[26]. BAI1 interacts with apoptotic myoblasts to transmit intracellular signals that promote myoblast fusion[21]. We demonstrated that uncoupling BAI3 from binding to Elmo blocks myoblast fusion[20]. Secreted C1q-Like 1–4 (C1qL1–4; CTRPs[27,28]) proteins are the only described ligands for BAI3[29]. Interplay between C1qLs and BAI3 was reported to regulate neuronal synapse formation[30–32]. While Elmo-binding and Rac1 signaling mediated by BAI3 are essential to promote fusion, whether this GPCR is capable of activating heterotrimeric G-proteins, and if this contributes to myoblast fusion, is unknown. One critical step toward answering this question is the identification of the molecules that control BAI3 activity in cell fusion.

We report here that BAI3-interacting proteins C1qL4 and Stabilin-2 act, respectively, as negative and positive regulators of BAI3 during myoblast fusion. Mixed populations cell fusion assays revealed that BAI3 and Stabilin-2 are both required on the same myoblast to promote fusion. Finally, we found that Stabilin-2 promotes myoblast fusion in part by activating the canonical GPCR activity of BAI3 which contributes to recruit Elmo proteins to the membrane where they can interact with BAI3. Our data suggest that the balance between inhibitory and activating proteins binding to BAI3 provide a tight control of myoblast fusion.

## Results

**C1qL1–4 proteins negatively regulates myoblasts fusion.** We identified BAI3 as a cell surface protein promoting myoblast fusion[20]. We aimed to determine here whether *Bai3* contributes to myogenesis in mice. Cross-sectional area (CSA) measurements revealed that 3-months-old *Bai3* knock-out animals display smaller fibers in the Tibialis Anterior (TA) compared to wild-type mice (Fig. 1a–c). Quantification of the numbers of nuclei located inside of the laminin-stained basement membrane and of Pax7-positive cells revealed a myonuclear number reduction for the Bai3-null mice, demonstrating that the reduced CSA is the result of a decrease in myoblast fusion (Fig. 1d–f). To further define the contribution of BAI3 to myogenesis, we tested its requirement during muscle regeneration. Following cardiotoxin (CTX)-induced injury to the TA, we found that BAI3-null animals showed less efficient muscle regeneration since smaller fibers could be observed in comparison to control animals (Fig. 1g–i). These experiments suggest that BAI3 contributes to myoblast fusion in vivo.

While expression of the pro-fusion proteins Myomaker, Myomixer and Stabilin-2 is upregulated at the time of fusion, BAI3 is also expressed in progenitors[12,15,19]. This prompted us to explore the mechanisms involved in regulating BAI3 activity. We hypothesized that pro-fusion activity of BAI3 is controlled by negative and positive regulators. We first explored the sole identified ligands of BAI3, C1qL1–4[29], as candidates to modulate fusion. We conducted gene expression analyses to establish whether C1qL-4 members are expressed in C2C12 cells, a well characterized in vitro model of mouse myoblast differentiation and fusion. *C1qL4* mRNA expression was detected during myoblast proliferation, but its expression decreased upon differentiation (Supplementary Fig. 1a, b). mRNA transcripts of *C1qL1*, *C1qL2*, and *C1qL3* mRNAs were expressed at low levels and not modulated during differentiation (Supplementary Fig. 1a, b). Hence, the expression pattern of C1qL4 suggest that it is unlikely to act as a positive signal for myoblast fusion.

To test whether C1qL4 could be a negative regulator of myoblast fusion, we depleted C1qL4 in C2C12 cells through shRNA expression. C1qL4 depletion significantly increased the fusion index, i.e., number of nuclei per Myosin Heavy Chain (MyHC)-positive myotube of three nuclei and more, which reflected increased fusion after 48 h of differentiation (Fig. 2a, b). The efficiency of *C1qL4* mRNA knockdown was confirmed by real-time quantitative PCR (Q-RT-PCR) (Fig. 2c). This observation could be reproduced using an independent shRNA targeting *C1qL4* (Supplementary Fig. 1c–e). The loss of C1qL4 function in the Sol8 myoblast cell line also increased the efficiency of myoblast fusion (Supplementary Fig. 1f–h). We monitored the expression of MyoD, Myogenin and Troponin T during differentiation, and we could demonstrate that their protein or mRNA levels are not affected by the knockdown of *C1qL4*, hence ruling out that depletion of C1qL4 facilitates the establishment of the differentiation program (Supplementary Fig. 2a–c). We did observe an increase in *MyHC* mRNA expression at days 2–4 of differentiation which is consistent with increased fusion, but expression of *Myomaker or Myomixer* was unchanged (Supplementary Fig. 2d–f). We found that the fusion index of control and C1qL4-depleted cells is similar at 24 and 36 h post-differentiation and that the increase in fusion in response to C1qL4 depletion is observable starting at 48 h (Supplementary Fig. 2g, h). Notably, the control C2C12 cells are unable to reach the fusion rate of the C1qL4-depleted cells even after 72 h of differentiation. We also confirmed that the bigger fibers observed when C1qL4 is depleted is due to an increase in fusion and not hypertrophy. To reach this

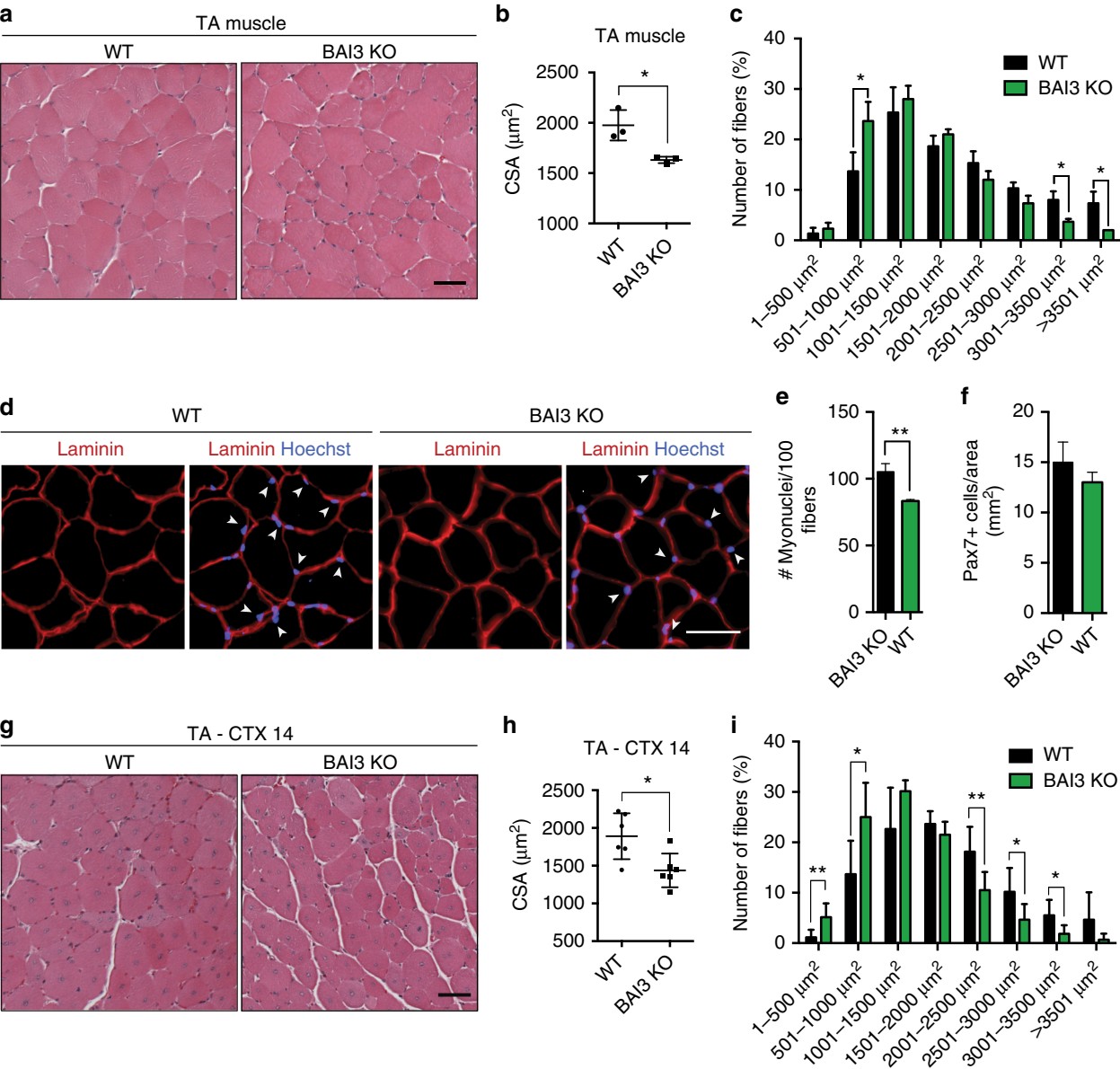

**Fig. 1** BAI3 KO mice display smaller myofibers and a less efficient muscle regeneration. **a** BAI3 KO adult mice display smaller myofibers than control animals. Representative H&E staining of the TA muscle section of BAI3 KO and WT mice. **b** Quantification of **a** showing the mean cross-sectional area (CSA) per mice ($n = 3$ mice). **c** Distribution of myofibers size of **a**. **d**, **e** Myonuclei number per fiber is reduced in BAI3 KO mice. **d** Cross-sectional muscle sections of BAI3 KO and control mice stained with anti-laminin (red) and Hoechst (blue). Arrowheads indicate myonuclei located inside the myofibers. **e** Quantification of the number of myonuclei located inside the myofibers of **d** ($n = 3$ mice). **f** Quantification of Pax7-positive cells of cross-sectional muscle sections (BAI3 KO and WT mice) stained with anti-Pax7. **g** BAI3 KO adult mice display smaller myofibers following cardiotoxin-induced injury of the TA muscle after 14 days of regeneration (H&E staining). **h** Quantification of **g** showing the mean CSA per mice ($n = 6$ mice). **i** Distribution of myofibers size of **g**. Error bars indicate standard deviation. Scale bar = 50 μm. One-way ANOVA followed by a Bonferroni test was used to calculate the $P$ values; *$P < 0.05$, **$P < 0.01$

conclusion, we measured the ratio of the total cell cytoplasm to nuclei area. These analyses revealed near threefold increase in total cell area upon depletion of C1qL4 with no effect on the cytoplasm to nuclei ratio indicating increased fusion (Supplementary Fig. 2i, j).

We used a gain-of-function approach to confirm the inhibitory role of C1qL1–4 in fusion. We overexpressed HA-alone or HA-tagged C1qL4 in C2C12 cells and found that HA-C1qL4 led to a block in fusion (Fig. 2d, e). We next treated C2C12 cells with either 100 ng/mL of bacterially expressed and purified GST or recombinant C1q domain of C1qL4 and found that the C1qL4

C1q domain blocked myoblast fusion (Fig. 2f, g and Supplementary Fig. 3a). We next assessed whether other members of the C1qL-family can also negatively regulate fusion. We transfected HEK293T cells to generate serum-free-conditioned media (CM) containing secreted HA-tagged full length C1qL1, C1qL2, C1qL3, or C1qL4 (Supplementary Fig. 3b, c). When the differentiation-conditioned media (10% supernatant containing C1qL proteins supplement with 2% horse serum in D-MEM) were added to C2C12 cells, we found that they all prevented myoblast fusion (Supplementary Fig. 3d, e). Finally, we isolated primary myoblasts from wild-type mice and treated them with 100 ng/mL of

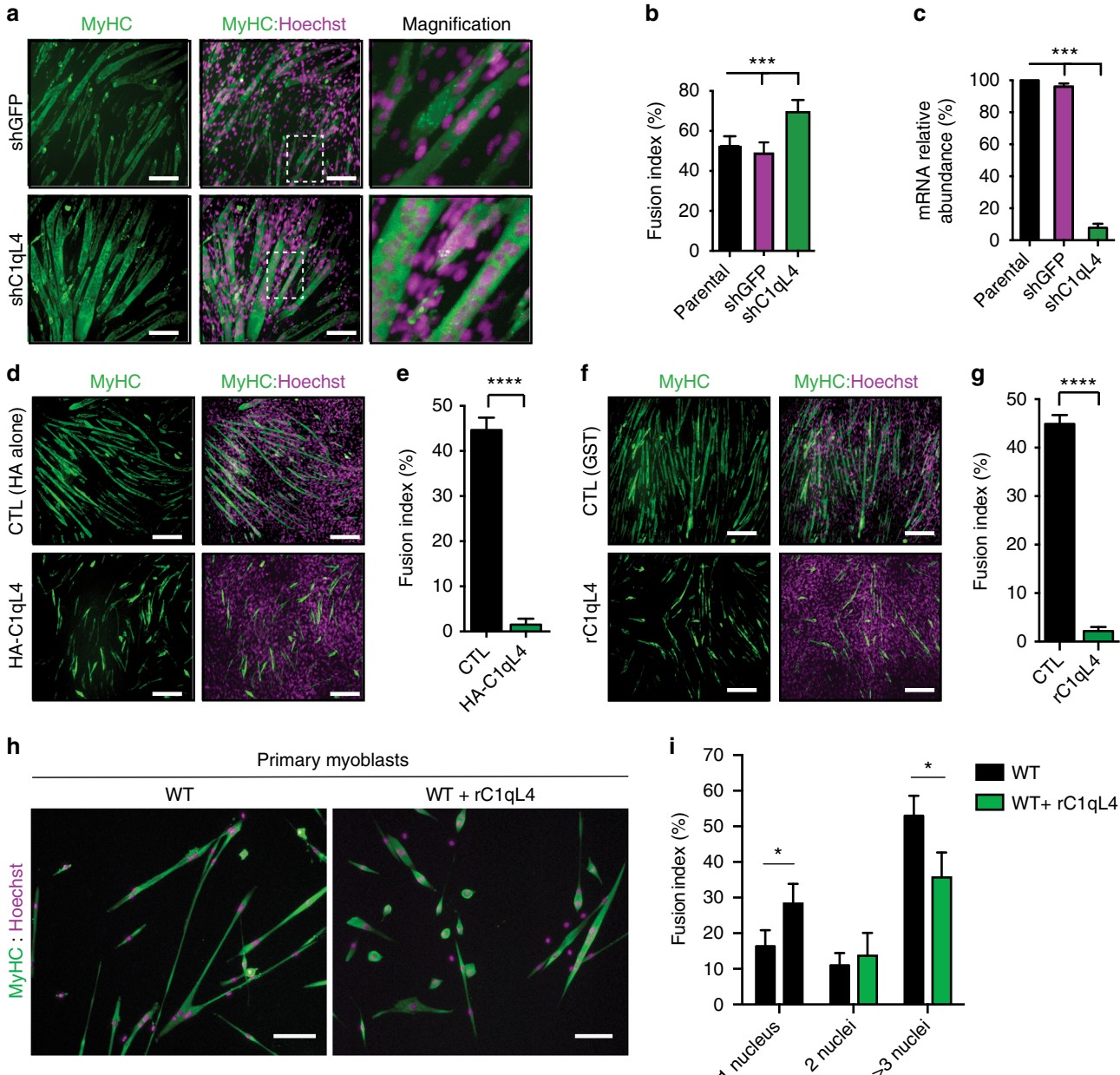

**Fig. 2** C1q-Like proteins negatively regulate myoblast fusion. **a** Downregulation of *C1qL4* increases myoblast fusion after 48 h of differentiation. Dotted white boxes are shown at higher magnification in the right panels. **b** To quantify myoblast fusion, MyHC-positive cells with three nuclei or more were considered as multinucleated myofibers and the fusion index was calculated by dividing the number of nuclei in multinucleated fibers by the total number of nuclei for each described condition. Quantification of experiment shown in **a**. **c** Real-time Q-RT-PCR amplifications of *C1ql4* were performed to confirm the specific shRNA-mediated knockdown. **d–g** Plasmid-mediated expression of HA-C1qL4, or addition of recombinant C1q domain of C1qL4 (rC1qL4), blocks C2C12 myoblast fusion. **d**, **f** Representative images of myoblast fusion from the indicated condition. **e**, **g** Quantification of experiments shown in **d** and **f**. **h** Addition of rC1qL4 blocks myoblast fusion of primary mouse myoblasts (72 h differentiation). **i** Quantification of experiments shown in **h**. Myofibers were stained for Myosin Heavy Chain (MyHC, MF20 antibody (green)) and nuclei were revealed by Hoechst (purple). Error bars indicate standard deviation. Scale bar = 100 μm (C2C12), scale bar = 50 μm (primary myoblasts). One-way ANOVA followed by a Bonferroni test was used to calculate the *P*-values; *$P < 0.05$, ***$P < 0.001$, ****$P < 0.0001$

recombinant C1q domain of C1qL4 and found that fusion is decreased (Fig. 2h, i). To confirm that C1qL4 is acting on fusion and not differentiation, we treated C2C12 cells with the recombinant C1q domain of C1qL4 and confirmed that the differentiation markers *MyoD*, *Myogenin*, and *MyHC4* mRNAs are expressed to levels similar to cells either untreated or treated with GST (Supplementary Fig. 3f). These results demonstrate that C1qL1–4 proteins are negative regulators of myoblast fusion.

**C1qL4 signals via BAI3 to negatively control myoblast fusion**. The C1qL1–4 proteins remain poorly characterized and may have additional targets than BAI3. To determine whether the inhibitory function of C1qL4 on myoblast fusion is specifically due to its interaction with BAI3, we generated a mutant of C1qL4 impaired for binding to BAI3. Exploiting the crystal structure of C1qL1, a BAI3-binding-deficient mutant of C1qL1 was previously generated by introducing two N-glycosylation sites in its C1q

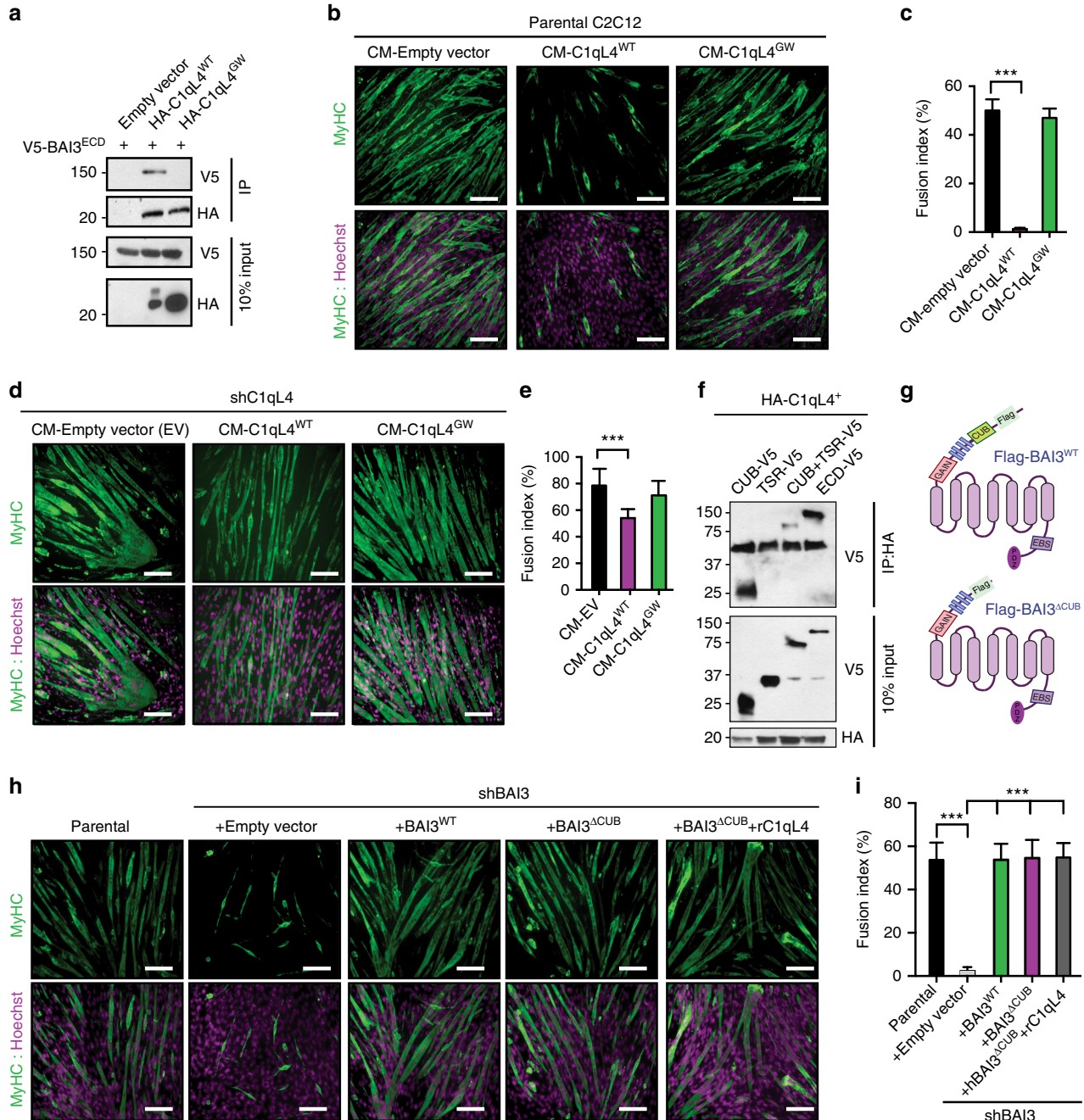

**Fig. 3** C1qL4 acts as a BAI3 ligand to negatively control myoblast fusion. **a** Generation of a C1qL4 mutant (C1qL4$^{GW}$) unable to bind BAI3. Anti-HA immunoprecipitations (IP) of HA-C1qL4$^{WT}$ or HA-C1qL4$^{GW}$ was carried out to assess the interaction with V5-BAI3$^{ECD}$. The HA-C1qL4$^{GW}$ mutant is defective to interact with V5-BAI3$^{ECD}$. **b** Treatment of C2C12 cells with C1qL4$^{WT}$-conditioned media (CM) blocks myoblast fusion. C1qL4$^{GW}$-conditioned media had no effect on myoblast fusion. **c** Quantification of experiments shown in **b**. **d** C2C12 cells expressing shC1ql4 display normal myoblast fusion when treated with HA-C1qL4$^{WT}$-conditioned media. HA-C1qL4$^{GW}$ was unable to revert the increased fusion following C1qL4 depletion. **e** Quantification of experiments shown in **d**. **f** The CUB domain of BAI3 is the minimal region necessary for interaction with C1qL4. Immunoprecipitations of HA-C1qL4 were carried out to assess which part of the BAI3 extracellular domains is responsible for the interaction with the ligand. **g** Schematic representation of Flag-BAI3$^{WT}$ and mutant (Flag-BAI3$^{\Delta CUB}$) where the CUB domain was deleted. **h** The CUB domain is dispensable for myoblast fusion. Expression of either BAI3$^{WT}$ or BAI3$^{\Delta CUB}$ restores fusion in C2C12 cells depleted of endogenous BAI3 by shRNA. **i** Quantification of experiments shown in **h**. Myofibers were stained for Myosin Heavy Chain (MyHC, MF20 antibody (green)) and nuclei were revealed by Hoechst (purple). Error bars indicate standard deviation. Scale bar = 100 μm. One-way ANOVA followed by a Bonferroni test was used to calculate the P-values; ***$P < 0.001$

domain which abolished C1qL1/BAI3 interactions[30]. We engineered two synonymous N-linked glycosylation sites at amino acid 172 and 203 in the C1q domain of C1qL4 (glycan wedge mutant: C1qL4$^{GW}$). We biochemically confirmed the loss of interaction between C1qL4$^{GW}$ and the BAI3 extracellular domain

(ECD) (Fig. 3a). To determine whether the BAI3-binding activity of C1qL4 is required to inhibit myoblast fusion, we added either HA-C1qL4$^{WT}$ or HA-C1qL4$^{GW}$ CM to the differentiation media of C2C12 cells and conducted a differentiation assay. While HA-C1qL4$^{WT}$ blocked myoblast fusion, the HA-C1qL4$^{GW}$ mutant

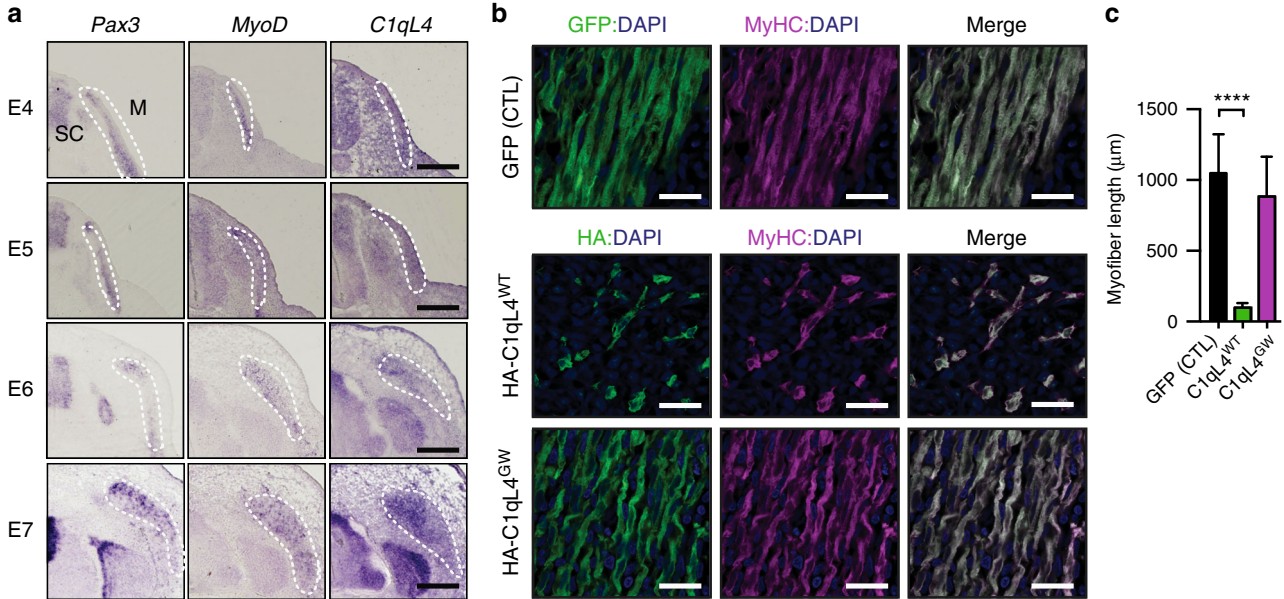

**Fig. 4** Overexpression of C1qL4 blocks myoblast fusion in ovo. **a** In situ hybridizations of anti-sense DIG-labeled riboprobes demonstrate that endogenous C1qL4 (right) is expressed in muscle cells that also express the myocyte differentiations Pax3 and MyoD (left and middle) at the indicated embryonic days of chicken embryo development. C1qL4 is also expressed in the spinal cord. SC, spinal cord; M, muscle. **b** Expression of HA-C1qL4$^{WT}$ in muscle progenitors blocks myoblast fusion in ovo in chick embryos. In contrast, C1qL4 deficient in BAI3-binding (HA-C1ql4$^{GW}$) has no effect on myoblast fusion. **c** Length of myofibers ($\mu$m) was quantified for each electroporation condition. Myofibers were stained for Myosin Heavy Chain (MyHC, MF20 antibody (purple)) and nuclei were revealed by DAPI (blue). Error bars indicate standard deviation. Scale bar = 100 $\mu$m. One-way ANOVA followed by a Bonferroni test was used to calculate the $P$-values; ****$P < 0.0001$

had no impact (Fig. 3b, c). We attempted to rescue this increase in fusion by adding exogenous HA-C1qL4$^{WT}$ or HA-C1qL4$^{GW}$ to C2C12 subjected to shRNA-mediated depletion of C1qL4. While addition of the CM-C1qL4$^{WT}$ restored a normal fusion rate (~50%), the CM-C1qL4$^{GW}$ had no functional impact (Fig. 3d, e). These results underline the importance of C1qL4 binding to BAI3 to inhibit fusion.

To further confirm that C1qL4 is functioning as a BAI3 ligand to inhibit fusion, we designed a mutant of BAI3 deficient in C1qL4-binding. Although the exact mechanism whereby C1qL-family proteins interact with BAI3 remain unclear, one study pointed to a key role for the TSRs for binding C1qL3[29] while another one highlighted the CUB domain to be responsible for binding C1qL1[30]. To determine the mechanism of the C1qL4/BAI3 interaction, we generated soluble fragments of each domain of the BAI3 extracellular portion and tested their ability to interact with HA-C1qL4. These experiments revealed that the CUB of BAI3 is the minimal and essential region carrying C1qL4-binding activity (Fig. 3f). To directly assess whether C1qL4 binding to BAI3 inhibits myoblast fusion, we generated a BAI3 mutant that is devoid of C1qL4-binding activity (Flag-BAI3$^{\Delta CUB}$) (Fig. 3g). We found that the deletion of the CUB domain does not alter the ability of this mutant to reach the cell surface (Supplementary Fig. 4a). We next validated the loss of binding of C1qL4 to the Flag-BAI3$^{\Delta CUB}$ in a cellular context. CM containing HA-C1qL4 was incubated with C2C12 cells expressing either Flag-BAI3$^{WT}$ or Flag-BAI3$^{\Delta CUB}$, and cells were washed and fixed prior to assess C1qL4/BAI3 interaction at the cell surface. While HA-C1qL4 decorated cells expressing Flag-BAI3$^{WT}$, it failed to bind to cells expressing Flag-BAI3$^{\Delta CUB}$ (Supplementary Fig. 4b). Re-expression of Flag-BAI3$^{\Delta CUB}$ in BAI3-depleted C2C12 cells revealed that the CUB domain is dispensable for myoblast fusion (Fig. 3h, i). While addition of recombinant C1qL4 inhibited myoblast fusion in cells expressing Flag-BAI3$^{WT}$ (Supplementary Fig. 4c, d), it failed to inhibit fusion

in cells overexpressing Flag-BAI3$^{\Delta CUB}$ (Fig. 3h, i). These results define the mechanism of interaction between C1qL4 and BAI3, and demonstrate that this coupling is essential to block the myoblast fusion activity of BAI3.

To determine the in vivo relevance of these findings, we turned to a gene expression approach in developing muscles of the chicken embryo. We carried out in situ hybridization assays with anti-sense digoxigenin-labeled riboprobes targeting *Pax3*, *MyoD*, and *C1qL4* on consecutive sections of embryos to determine whether *C1qL4* is expressed in muscles. We found that during embryonic day 4 to 7, developmental stages when the first wave of myogenesis occurs to form multinucleated fibers, *C1qL4* is expressed by muscle cells positive for the myocyte differentiation markers *Pax3* and *MyoD* (Fig. 4a). We detected expression of *C1qL4* in the developing spinal cord (Fig. 4a), as previously reported[33]. We next forced the expression of either GFP alone, HA-C1qL4$^{WT}$ or HA-C1qL4$^{GW}$ in developing muscles by somite electroporation[20,34,35]. While expression of GFP did not affect myoblast fusion in vivo, exogenous HA-C1qL4$^{WT}$ inhibited fusion and HA-C1qL4$^{GW}$ had no effect (Fig. 4b, c). Expression of HA-C1qL4 did not interfere with differentiation as evaluated by expression of *Myogenin* (Supplementary Fig. 4e, f). These data identify C1qL4 binding to BAI3 as a spatiotemporal negative signal of myoblast fusion.

**Stabilin-2 is a BAI3-interacting transmembrane protein**. We next searched for proteins that could be responsible for activating BAI3. We first investigated whether the extracellular region of BAI3 was required for cell fusion. We found that a BAI3 mutant lacking the extracellular region, BAI3$^{\Delta N}$, blocked fusion when expressed in myoblasts (Fig. 5a, b). In contrast to BAI3$^{WT}$, BAI3$^{\Delta N}$ was unable to rescue fusion in C2C12 depleted of endogenous BAI3 (Fig. 5c, d). We reasoned that if the extra-cellular portion of BAI3 is binding a protein essential for

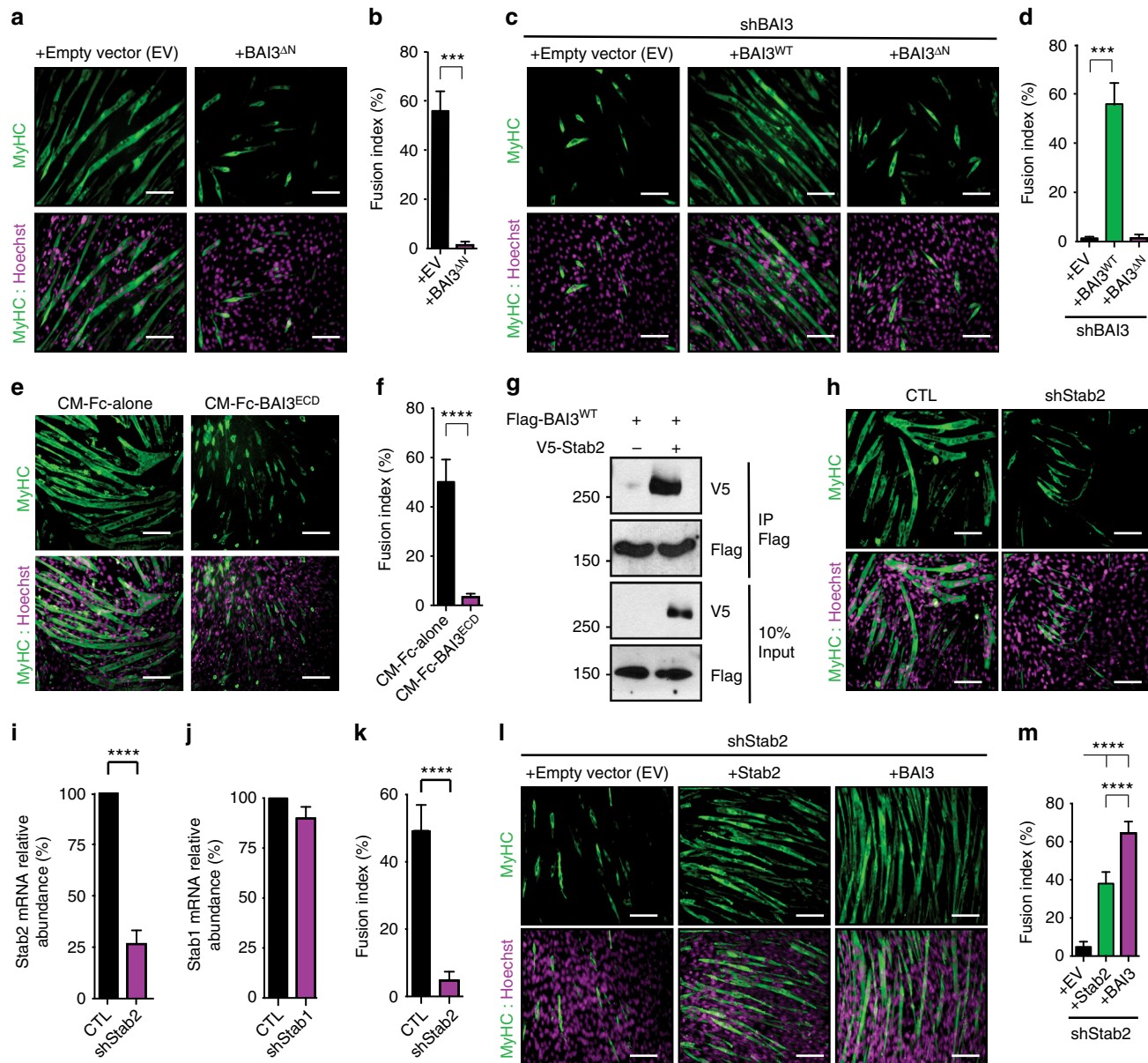

**Fig. 5** Stabilin-2 is a binding partner of BAI3. **a** A BAI3 mutant lacking the extracellular domain (BAI3$^{\Delta N}$) is defective to promote myoblast fusion. Expression of Flag-BAI3$^{\Delta N}$ blocks myoblast fusion in parental C2C12 cells. **b** Quantification of experiments shown in **a**. **c** Re-expression of Flag-BAI3$^{\Delta N}$ is unable to restore myoblast fusion in C2C12 cells depleted of endogenous BAI3. **d** Quantification of experiments shown in **c**. **e** The BAI3$^{ECD}$ acts as a decoy receptor. C2C12 treated with Fc-BAI3$^{ECD}$-conditioned media, but not Fc-alone, display a block in myoblast fusion. **f** Quantification of the experiment shown in **e**. **g** Immunoprecipitation of Flag-BAI3 was carried out to assess its interaction with full length V5-Stabilin-2. **h**–**k** C2C12 cells expressing GFP or shRNA targeting Stabilin-2 were generated by retroviral infections. **h** Downregulation of Stabilin-2 decreases myoblast fusion after 48 h differentiation condition. **i**, **j** Real-time Q-RT-PCR amplifications of Stabilin-1 and Stabilin-2 were performed to confirm the specificity of the knockdowns. **k** Quantification of experiments shown in **h**. **l** Fusion of C2C12 cells depleted of Stabilin-2 (shStab2) is restored following expression of either human V5-Stabilin-2 or Flag-BAI3. **m** Quantification of experiments shown in **l**. Myofibers were stained for Myosin Heavy Chain (MyHC, MF20 antibody (green)) and nuclei were revealed by Hoechst (purple). Error bars indicate standard deviation. Scale bar = 100 μm. One-way ANOVA followed by a Bonferroni test was used to calculate the *P*-values; ****P* < 0.001, *****P* < 0.0001

differentiation, adding a recombinant and soluble portion of the extracellular part of BAI3 should act as a decoy receptor. We therefore differentiated C2C12 cells in the presence of conditioned media enriched with either Fc alone or the soluble extracellular region of BAI3 (Fc-BAI3$^{ECD}$). Myoblast fusion was impaired in the presence of Fc-BAI3$^{ECD}$ in comparison to the control Fc-alone condition (Fig. 5e, f). To identify the functional region in the extracellular portion of BAI3 responsible for trapping the putative binding protein, we conducted a similar experiment where the BAI3$^{ECD}$ was further fragmented into

domains: CUB, TSRs, and the CUB-TSRs chimera. We found that the addition of the TSRs domain was the minimal region of BAI3 sufficient to inhibit myoblast fusion (Supplementary Fig. 5a, b). These results suggest that C2C12 express a BAI3-interacting protein that promotes myoblast fusion.

To identify such candidate BAI3-binding proteins, we purified Fc-alone or Fc-BAI3$^{ECD}$ from transfected HEK293T cells and used them to affinity purify proteins from the supernatant of differentiating C2C12 (pooled from $t = 0$, 24, 48 h). Bound proteins were identified by mass spectrometry (Supplementary

Fig. 6a). Components of the extracellular matrix (Tenascin C, Biglycan, Periostin, Midkine) were identified in the Fc-BAI3[ECD] purification (Supplementary Fig. 6b). A candidate receptor protein, Stabilin-2 (Supplementary Fig. 6a, b), caught our attention since it is a regulator of myoblast fusion[18,19]. The low number of peptides for Stabilin-2 likely reflects the purification of a poorly abundant cleaved fragment of this receptor[36]. C2C12 cells were transfected with V5-Stabilin-2 and Flag-BAI3[ECD], and we carried out a co-immunoprecipitation assay that demonstrated the binding of the two proteins (Supplementary Fig. 6c). We validated the binding between full-length Flag-BAI3 and V5-Stabilin-2 (Fig. 5g). To further dissect the interaction between Stabilin-2 and BAI3, we conducted a co-immunoprecipitation assay that revealed that Stabilin-2 is unable to bind BAI3[ΔN] (Supplementary Fig. 6d). We conducted a proximity ligation assay (PLA) to reveal the interaction of Flag-BAI3 with endogenous Stabilin-2. We expressed tracer amounts of Flag-BAI3 in C2C12 cells and allowed them to differentiate for 24 h. We performed the PLA assay using anti-Flag and anti-Stabilin-2 antibodies, which generated a specific PLA signal, indicating that these two cells surface proteins are in close proximity at the surface of myoblasts (Supplementary Fig. 6e). We depleted Stabilin-2 by shRNA in C2C12 cells and conducted differentiation assays that confirmed the previously reported function of Stabilin-2 as a promoter of myoblast fusion (Fig. 5h–k). We found that the defect in myoblast fusion in cells where Stabilin-2 was depleted by shRNA could be rescued by expression of either V5-Stabilin-2 or Flag-BAI3 (Fig. 5l, m). Depletion of either BAI3 or Stabilin-2 did not impair myoblast fusion by preventing the expression of *Myomaker* or *Myomixer* (Supplementary Fig. 6f, g). This suggest that Stabilin-2 physically and functionally interacts with BAI3 during myoblast fusion.

**BAI3 and Stabilin-2 interact in cis to promote cell fusion**. We hypothesized that BAI3 and Stabilin-2 of two fusing myoblasts may functionally interact either in cis (both receptors on the same cell) or in trans (one receptor on each cell). To address this, we developed a mixed myoblast population assay by co-incubating at a 1:1 ratio different populations of myoblasts, pre-stained with the general membrane dyes PKH26-GL (red; pseudo colored purple) or PHK67 (green). As a positive control, we found that when green and purple parental C2C12 myoblasts were mixed, myotubes formed efficiently and were double positive for green and purple (Fig. 6a). If we mixed parental myoblasts with either myoblasts overexpressing Flag-BAI3 or V5-Stabilin-2, the resulting myotubes were composed of the two cell populations (Fig. 6b, c). In contrast, when green C2C12 depleted of BAI3 were mixed with purple C2C12 overexpressing V5-Stabilin-2, the myotubes that formed were mainly purple and the green C2C12 depleted of BAI3 remained mononucleated (Fig. 6d). When purple C2C12 depleted of Stabilin-2 were mixed with green C2C12 overexpressing Flag-BAI3, the myotubes that formed were mainly green and the purple C2C12 depleted of Stabilin-2 remained mononucleated (Fig. 6e). Finally, we mixed C2C12 depleted of BAI3 with C2C12 depleted of Stabilin-2 and observed that all cells remained mononucleated (Fig. 6f).

We next investigated whether the fusion promoting activity of Stabilin-2 and BAI3 is restricted to myoblast-myoblast fusion or if it also contributes to myoblast–myotube fusion. The mixed population assay was modified to have myoblast added to preformed myotubes. When green parental cells were mixed with purple myotubes, we observed the presence of double positive green and purple myotubes (Supplementary Fig. 7a). If we mixed parental green myoblasts with purple fibers overexpressing BAI3 or Stabilin-2, double positive green-purple fibers were observed

(Supplementary Fig. 7b, c). When green myoblasts depleted of either BAI3 or Stabilin-2 were mixed with purple parental myotubes, we noted that these cells failed to fuse with the myotubes (Supplementary Fig. 7d, e). After mixing green cells lacking BAI3 or Stabilin-2 with purple myotubes overexpressing one of the receptors, no double positive fusion events could be observed (Supplementary Fig. 7f, g). These results suggest that BAI3 and Stabilin-2 act in cis during myoblast/myotube fusion.

**Auto-proteolysis of BAI3 is not required for myoblast fusion**. While interaction of BAI3 and Elmo is essential for myoblast fusion[20], whether intrinsic GPCR activity is present and is also required in this biological context is unexplored. Most Adhesion GPCRs require cleavage to reveal their activity[24,25], including BAI1[37–40]. To investigate whether BAI3 undergoes GAIN-mediated cleavage, we generated a mutant of the GAIN domain by replacing a conserved arginine essential for cleavage in other GAIN domains[25] (BAI3[R836A]) (Supplementary Fig. 8a). If BAI3 undergoes GAIN-mediated cleavage, this should generate a soluble fragment of approximately 120 kDa. When Flag-tagged BAI3[WT] or BAI3[R836A] were overexpressed in HEK293T, we failed to detect a cleaved fragment either in the supernatants (enriched by Flag immunoprecipitation) or in the total cell lysates (Supplementary Fig. 8b). We considered that auto-cleavage of BAI3 could require additional factors expressed during myoblast differentiation. We expressed Flag-BAI3 in C2C12 cells and analyzed whether BAI3 was cleaved post-differentiation (0–48 h) and found that it remained a full-length protein (Supplementary Fig. 8c). We next hypothesized that BAI3 may need to complex with Stabilin-2 to adopt a productive conformation for auto-cleavage. We found that expression of V5-Stabilin-2 with Flag-BAI3 did not induce cleavage of BAI3 (Supplementary Fig. 8d). To exclude the possibility that a small pool of BAI3 may be cleaved and important for myoblast fusion, we conducted rescue assays with a panel of BAI3 GAIN mutants in BAI3-depleted C2C12 cells (BAI3[R836A], BAI3[L837A], BAI3[S838A]; Supplementary Fig. 8a). We confirmed that these mutations did not impair their cell surface localization (Supplementary Fig. 8e). These BAI3 mutants rescued the myoblast fusion defect of BAI3-depleted C2C12 cells as efficiently as BAI3[WT] (Supplementary Fig. 8f, g). Collectively, these data suggest that BAI3 cleavage is not required for the receptor to promote myoblast fusion.

**G-proteins promote recruitment of Elmo at the membrane**. While it remains unknown whether BAI3 functions as a GPCR for heterotrimeric G-proteins, an engineered cleaved form of BAI1 displays GPCR activity[40]. A similar mutant of BAI3, V5-BAI3[ΔN] (Fig. 5a), was tested for its ability to couple to β-Arrestin2, a surrogate for GPCR signaling. This revealed that BAI3[ΔN], in contrast to BAI3[WT], co-immunoprecipitated with β-Arrestin (Fig. 7a). β-Arrestin co-localized with BAI3[ΔN] but not with V5-BAI3[WT], at the plasma membrane of C2C12 (Fig. 7b) and COS7 (Supplementary Fig. 9a) cells. We measured whether BAI3 can activate heterotrimeric G-proteins by Bioluminescence Resonance Energy Transfer 2 (BRET2). While the basal BRET2 signal from RlucII-tagged $G_{\alpha i1}$ and GFP10-tagged $G_{\gamma 2}$ was unaffected by expression of BAI3[WT], a decrease in the signal was observed upon expression of BAI3[ΔN] (Fig. 7c), indicative of BAI3-mediated activations of $G_{\alpha i1}$. As a positive control, we found that stimulation of CXCR4 with its ligand CXCL12 led to a decrease in BRET2, indicative of $G_{\alpha i1}$ and $G_{\beta \gamma}$ dissociation (Fig. 7c)[41]. We failed to detect GPCR activity in BAI3[ΔN] when additional RlucII-$G_{\alpha}$ subunits were tested in this BRET2 assay ($G_s$, $G_i$ $G_{12/13}$, $G_q$; Supplementary Fig. 9b–d). These experiments reveal that BAI3 displays regulated GPCR activity.

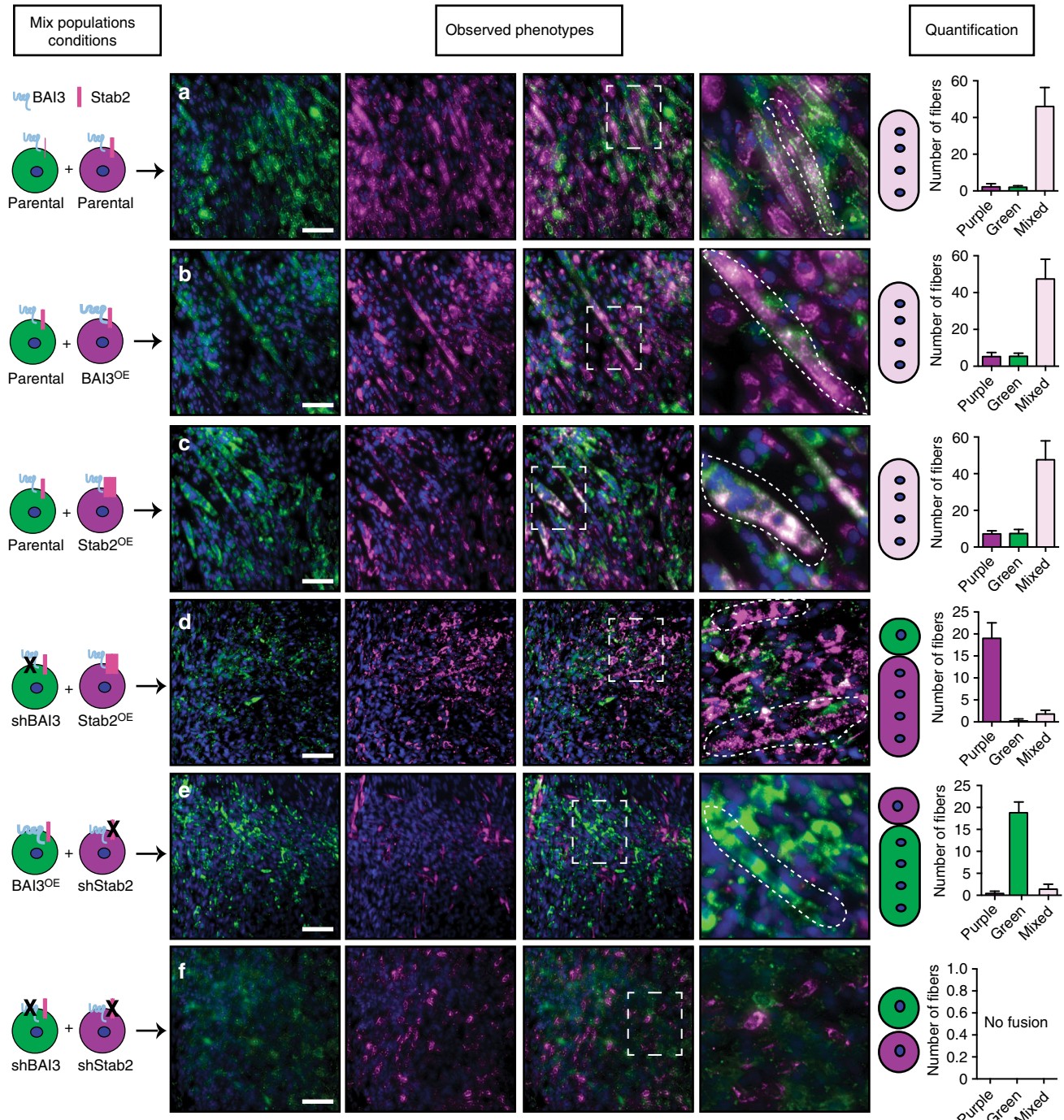

**Fig. 6** BAI3 and Stabilin-2 interact in cis in myoblasts to promote cell fusion. Stabilin-2 and BAI3 are required on the same myoblast to promote cell fusion. **a** Parental C2C12 cells were stained with membrane dyes and mixed in a 1:1 ratio. 24 h after plating, cells were switched into differentiation conditions for 48 h. **b** Parental C2C12 mixed with C2C12 overexpressing Flag-BAI3. **c** Parental C2C12 (green) mixed with C2C12 overexpressing V5-Stabilin-2 (purple). Cell fusion was intact and mixed fibers were observed after differentiation. **d** C2C12-expressing shRNA against BAI3 (green) are mixed with C2C12 overexpressing Stabilin-2 (purple). Cells without BAI3 were unable to fuse and mix with other population after differentiation. **e** C2C12 expressing shRNA against Stabilin-2 (purple) were mixed with C2C12 overexpressing BAI3 (green). Mixed fibers were not observed and only cells expressing both receptors were able to fuse together after differentiation. **f** Cells-expressing shRNA against Stabilin-2 or BAI3 were mixed and no fusion was observed following differentiation. The quantification of the phenotypes was analyzed from multiple images. Scale bar = 100 μm

Elmo proteins are effectors of $G_{\alpha i}$ and $G_{\beta \gamma}$[42,43]. BAI3-mediated activation of $G_{\alpha i}$ and $G_{\beta \gamma}$ could contribute to recruit Elmo to BAI3 to promote fusion. We tested whether Elmo2 can interact with $G_{\alpha i}$ or $G_{\beta \gamma}$ subunits in living cells. When we expressed Myc-Elmo2, we found that the basal BRET2 signal from RlucII-tagged $G_{\alpha i1}$ and GFP10-tagged $G_{\gamma 2}$ decreased (Fig. 7d), indicative that

Elmo2 interacts with G-proteins[42,44]. Elmo2 did not affect the coupling of $G_{\alpha i2}$, $G_{\alpha i3}$ or $G_{\alpha 12}$ to GFP10-$G_{\gamma 2}$ (Fig. 7d). We explored whether the GPCR activity of BAI3 may facilitate the recruitment of Elmo2 to the membrane. We found that Myc-Elmo2 was distributed in the cytosol when expressed alone or expressed with BAI3^WT in C2C12 (Fig. 7e, f) or COS7

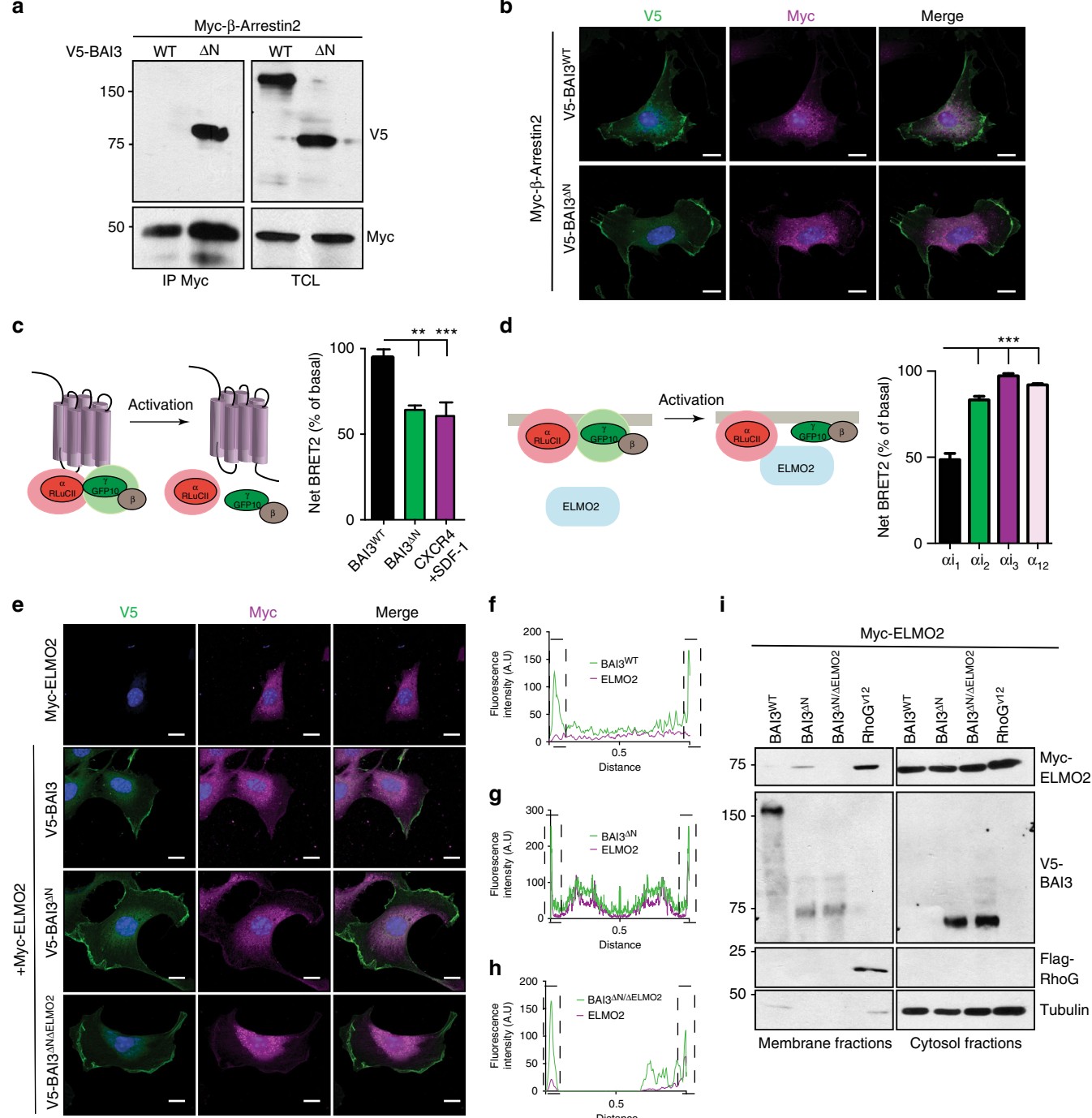

**Fig. 7** Canonical and non-canonical signaling from BAI3 are required to promote the recruitment of ELMO at the membrane. **a** A mutant of BAI3 lacking the extracellular domain (BAI3$^{\Delta N}$) robustly interacts with β-Arrestin2. Immunoprecipitations of Myc-β-Arrestin2 revealed an interaction with Flag-BAI3$^{\Delta N}$ but not with Flag-BAI3$^{WT}$. **b** Immunofluorescence experiments assessing the ability of BAI3$^{WT}$ or BAI3$^{\Delta N}$ (green) to recruit Myc-β-Arrestin2 (purple) at the membrane of C2C12 cells. **c** Effect of BAI3$^{WT}$ and BAI3$^{\Delta N}$ on BRET2 signal between G proteins G$_{\alpha i1}$-RlucII and GFP10-G$_{\gamma 2}$ in cells co-transfected with the untagged G$_{\beta}$. **d** Effect of ELMO2 on the BRET2 signal between G proteins G$_{\alpha i1}$-RlucII and GFP10-Gγ2 in cells co-transfected with the untagged G$_{\beta}$. Net BRET2 signal is normalized to basal BRET2 signal. **e–h** BAI3$^{\Delta N}$ promotes the recruitment of ELMO2 at the membrane in C2C12 cells. **e** Immunofluorescence assessing ELMO2 (purple) recruitment at the membrane in the presence of either Flag–tagged BAI3$^{WT}$, Flag-BAI3$^{\Delta N}$ or BAI3$^{\Delta N/}$$^{\Delta ELMO2}$ (green). **f–h** Quantification of experiment shown in **e**. **i** Membrane fractionation experiments of HEK293T cells followed by western blot analyses demonstrate the ability of BAI3$^{\Delta N}$ to recruit ELMO2 at the membrane. Error bars indicate standard deviation. Scale bar = 100 μm. One-way ANOVA followed by a Bonferroni test was used to calculate the $P$-values; **$P < 0.01$, ***$P < 0.001$

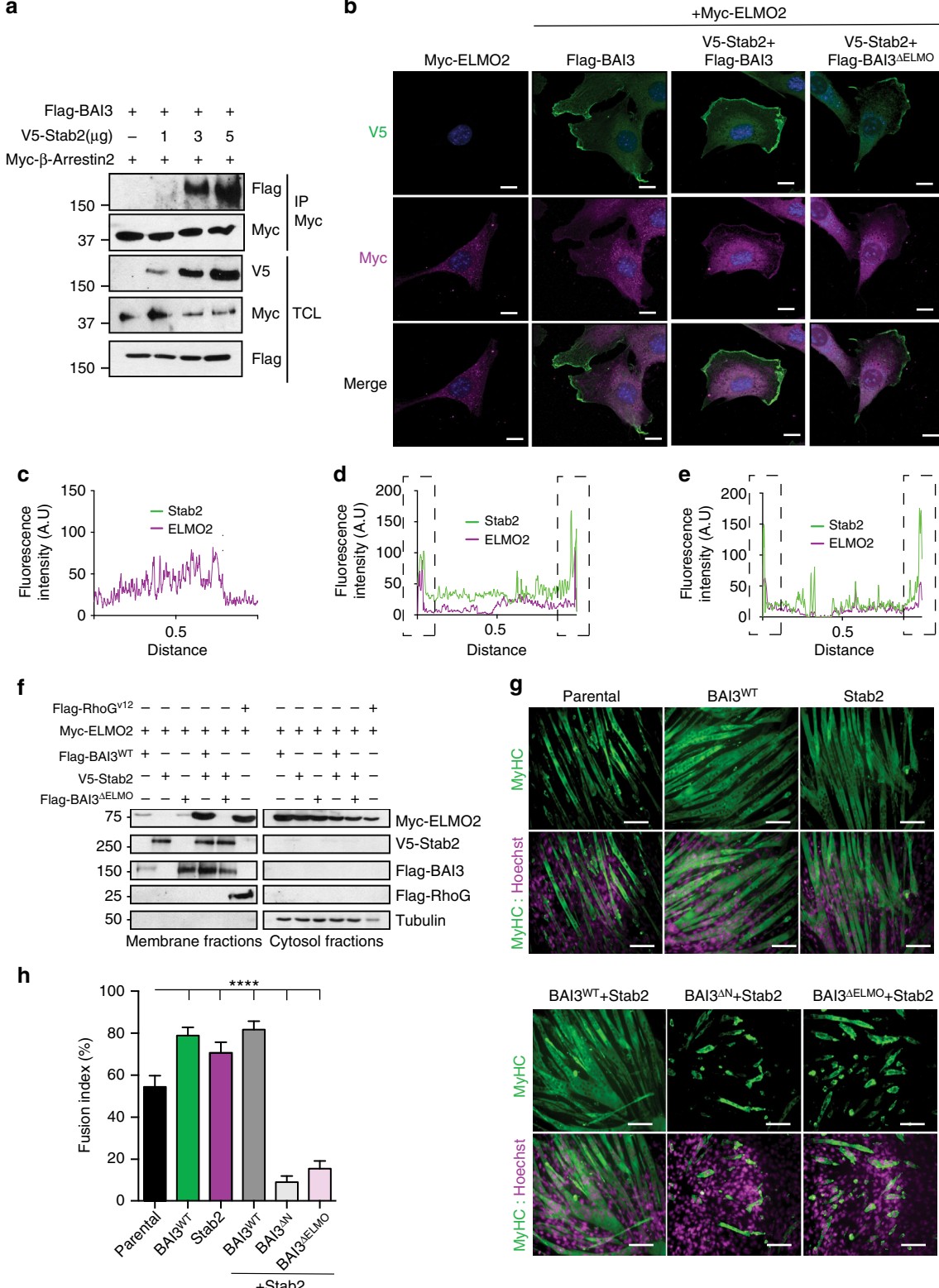

(Supplementary Fig. 9e, f) cells. However, BAI3$^{\Delta N}$ promoted the recruitment of Elmo2 at the membrane in C2C12 (Fig. 7e, g) and COS7 (Supplementary Fig. 9e, g) cells. To assess whether the recruitment of Elmo2 at the membrane was driven by the activation of G-proteins, or alternatively directly by the C-terminal Elmo-binding region of BAI3, we generated a BAI3$^{\Delta N}$ mutant lacking Elmo-binding activity (BAI3$^{\Delta N/\Delta ELMO}$). We found that BAI3$^{\Delta N/\Delta ELMO}$ was able to recruit Elmo2, albeit less efficiently than the wild receptor, to the membrane in C2C12 (Fig. 7e, h) or COS7 (Supplementary Fig. 9e, h) cells. These experiments revealed a contribution of the GPCR activity of BAI3 to recruit Elmo2 at the membrane. We next carried out a biochemical cell fractionation assay[45,46] to assess the mechanism leading to Elmo recruitment to the membrane. This revealed that

**Fig. 8** Stabilin-2 promotes BAI3 GPCR activity and facilitates the recruitment of ELMO2 at the membrane. **a** Stabilin-2 activates BAI3. Increasing amounts of Stabilin-2 lead to an increase in β-arrestin2 coupling to BAI3. **b** Immunofluorescence experiments assessing ELMO2 (purple) recruitment at the membrane of C2C12 cells in the presence of either Flag-BAI3$^{WT}$, BAI3$^{\Delta ELMO2}$ or V5-Stabilin-2 (green). **c–e** Quantification of experiment shown in **b**. **f** Membrane fractionation experiment followed by western blot analyses of HEK293t cells demonstrate that Stabilin-2 cooperates with BAI3 to promote the recruitment of ELMO2 at the membrane. BAI3$^{\Delta ELMO2}$ fails to recruit ELMO2 at the membrane in the presence of Stabilin-2. **g** The extracellular domain and the Elmo binding site of BAI3 are both required to promote myoblast fusion. C2C12 overexpressing either Flag-BAI3, V5-Stabilin-2 or both together exhibit an increase in fusion. Cells expressing Flag-BAI3$^{\Delta N}$ mutant together with V5-Stabilin-2 display a decrease in fusion. Similarly, the expression of BAI3$^{\Delta ELMO2}$ together with V5-Stabilin-2 blocked myoblast fusion. **h** Quantification of the experiment shown in **g**. Myofibers were stained for Myosin Heavy Chain (MyHC, MF20 antibody (green)) and nuclei were revealed by Hoechst (purple). Error bars indicate standard deviation. Scale bar = 100 μm. One-way ANOVA followed by a Bonferroni test was used to calculate the $P$-values; ****$P < 0.0001$

BAI3$^{\Delta N}$, similar to active RhoG, promoted the translocation of a pool of Myc-Elmo2 in the membrane fraction (Fig. 7i). BAI3 possesses an Elmo-binding site at its C-terminus that may also participate to recruit Elmo proteins to the membrane. To test this, we used the BAI3$^{\Delta N/\Delta ELMO}$ and found that it was incapable of promoting the translocation of Myc-Elmo2 to the membrane (Fig. 7i). These data demonstrate that BAI3-mediated activation of G-proteins promotes the initial recruitment of Elmo at the membrane and that the C-terminus of BAI3 is required for their anchoring to the membrane.

**Stabilin-2 promotes the GPCR activity of BAI3**. We next sought to determine whether the formation of a complex composed of Stabilin-2 and BAI3 could be a physiological signal to activate BAI3. We monitored the interaction of β-Arrestin2 with BAI3, a surrogate for activation of the receptor, upon expression of Stabilin-2. Co-immunoprecipitation assays demonstrated that V5-Stabilin-2 promoted the interaction of Myc-β-Arrestin2 with Flag-BAI3 (Fig. 8a). We investigated whether Stabilin-2-mediated activation of BAI3 GPCR activity is sufficient to recruit Elmo2 at the membrane. We found that Myc-Elmo2 was distributed in the cytosol when expressed alone or expressed with BAI3$^{WT}$ in C2C12 (Fig. 8b, c) and COS7 (Supplementary Fig. 10a, b) cells. Expression of Stabilin-2 with BAI3 promoted a recruitment of Elmo2 at the membrane in C2C12 (Fig. 8b–d) and COS7 (Supplementary Fig. 10a, c) cells. To assess whether the recruitment of Elmo2 at the membrane was driven by the activation of G-proteins, or alternatively by the C-terminal Elmo-binding region of BAI3, we expressed Stabilin-2 with BAI3$^{\Delta ELMO}$. The Stabilin-2/BAI3$^{\Delta ELMO}$ complex was able to recruit Elmo2 at the membrane, albeit less efficiently than the wild receptor in C2C12 (Fig. 8b–e) or COS7 (Supplementary Fig. 10a, d) cells. We carried out cell fractionation assays that revealed that Myc-Elmo2 was not recruited to the membrane fraction when Flag-BAI3$^{WT}$, Flag-BAI3$^{\Delta ELMO}$, or V5-Stabilin-2 were expressed (Fig. 8f). Co-expression of Stabilin-2 with BAI3$^{WT}$, but not BAI3$^{\Delta ELMO}$, led to a translocation of Elmo2 to the membrane (Fig. 8f). We tested the importance of the BAI3/Stabilin-2 complex for myoblast fusion. Expression of BAI3, Stabilin-2, or BAI3/Stabilin-2 increased myoblast fusion in comparison to control cells (Fig. 8g, h). Expression of Stabilin-2 with BAI3$^{\Delta N}$ (unable to bind Stabilin-2) impaired myoblast fusion (Fig. 8g, h). Likewise, expression of Stabilin-2 with BAI3$^{\Delta ELMO}$ impaired myoblast fusion. These data define that Stabilin-2 promotes the GPCR activity of BAI3 for recruitment of Elmo at the membrane and their anchoring to BAI3 via a direct interaction to promote myoblast fusion.

## Discussion

We identified BAI3 as a promoter of myoblast fusion via its signaling through Elmo/Dock[6,20]. BAI3, in contrasts to Myo-maker or Stabilin-2, is not regulated at the transcriptional level during differentiation which suggests that molecular mechanisms

must be in place to control its activity. Here, we identified the proteins that spatiotemporally control the activity of BAI3 during myoblast fusion. C1qL1–4 negatively regulate cell fusion by engaging BAI3 (model Fig. 9a, b). The pattern of expression of *C1qL4* that we describe in C2C12 myoblasts is consistent with this protein being expressed prior to the timing when myoblast fusion is deployed and that its expression is shut down at the time when fusion is taking place (Fig. 9c). The secreted C1qL1–4 may provide checkpoints that allow myoblast fusion to occur at precise times in vivo. In the developing muscles of the chick embryo, such a clear decrease in *C1qL4* expression was not observed. One explanation is that in vitro, myoblasts are synchronized to enter into differentiation simultaneously while in vivo this is less synchronized as differentiation occurs over several days of development.

A key finding of our study is the identification of Stabilin-2 as a BAI3-interactor (model Fig. 9c, d). Stabilin-1 and Stabilin-2 have broad biological roles as scavenger receptors (e.g., for oxidized low-density lipoproteins[36]) and phosphatidylserine receptors (e.g., for the clearance of apoptotic cells[47–49]). Atypical transient exposure of phosphatidylserine at the surface of fusogenic cells is a pro-fusion signal[50,51], and Stabilin-2 was found to be the missing receptor to transduce this signal[18,19]. Our results suggest that Stabilin-2 could transmit its pro-fusion signals in myoblasts via Elmo-Dock by interacting with BAI3. Similarly, Stabilin-2 promotes the engulfment of apoptotic cells by heterodimerizing with Integrin αvβ5 that signals via Dock1[47,52].

We demonstrate that BAI3 displays GPCR activity. An engineered truncation in BAI1, mimicking auto-cleavage, is activating[40]. We generated a similar mutant of BAI3 (BAI3$^{\Delta N}$) and used BRET2 biosensors to assay its ability to activate G-proteins. We found that BAI3 activates G$_{\alpha i1}$ (model Fig. 9d). We identified Stabilin-2, which is upregulated at the time of fusion[18,19], as a physiological signal to activate BAI3 (model Fig. 9c, d).

Myoblast fusion in *Drosophila* is driven by two distinct cell populations: the fusion competent myoblasts and the founder cells[2]. The existence of such populations has not yet been established in vertebrates. We wanted to understand the interaction mechanism between Stabilin-2 and BAI3 on the fusing myoblasts. We hypothesized two scenarios: (1) both receptors could be required on the same myoblast (in cis) or (2) each receptor could be asymmetrically distributed on each fusing cell (in trans). Our results revealed that the Stabilin-2/BAI3 complex functions in cis to promote fusion and that both fusing myoblasts, or myoblast-myotube, require both receptors on their surface (model Fig. 9d, e). These observations are in line with one population of myoblasts fusing together in vertebrates.

BAI1–3 have an Elmo-binding site at their C-terminus that binds Elmo in vitro[20,23]. Our assumption was that Elmo proteins would co-localize with BAI3, but this was not observed in cells. This suggests that the Elmo-binding site on BAI3 is "masked" at the basal state. We found that artificial (expression of BAI3$^{\Delta N}$) or physiological (co-expression with Stabilin-2) activation of BAI3

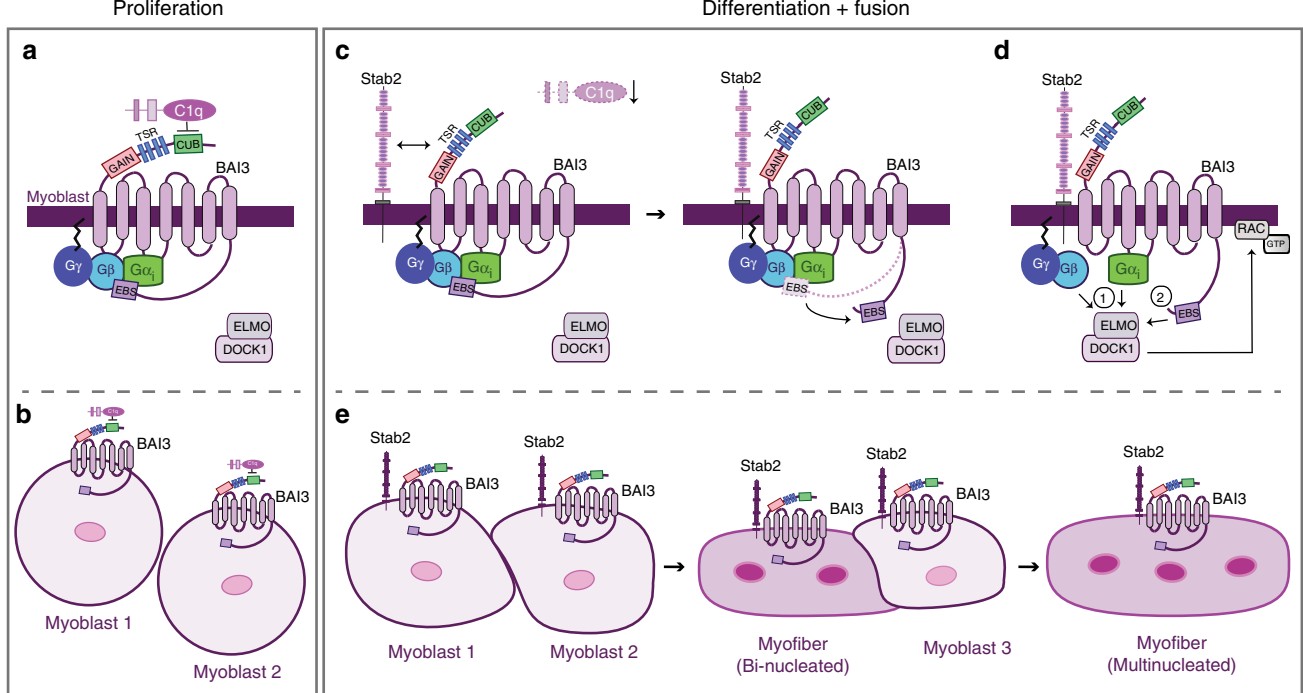

**Fig. 9** Working model. **a, b** In muscle progenitors, BAI3 activity is inhibited by the interaction of the secreted protein C1qL4 binding to the CUB domain found in the extracellular region of BAI3. **c** When differentiation is initiated, the expression of C1qL4 decreases while the expression of Stabilin-2 increases. The interaction between BAI3 and Stabilin-2 leads to the activation of the canonical signaling of G protein and the exposure of the ELMO Binding site (EBS) of BAI3. **d** These events lead to the recruitment of the ELMO/Dock1 complex by two signals: 1-the G proteins broadly recruit ELMO/DOCK1 to the membrane and 2-this recruitment facilitates a direct interaction of ELMO with BAI3 via a direct interaction of ELMO with the EBS of BAI3. These interactions collectively promote myoblast fusion. **e** The mixed population fusion assays conducted in this study demonstrated that both BAI3 and Stabilin-2 receptors are required on both fusing cells for both myoblast–myoblast and myoblast–myotube fusion

leads to Elmo translocation to the membrane and co-localization with BAI3 proteins. Our results revealed that the recruitment of Elmo proteins to BAI3 occurs in a two-step manner (model Fig. 9c, d). First, BAI3-mediated activation of G-proteins provides a signal to recruit a pool of Elmo at the membrane. We reached this conclusion by using mutants of BAI3 lacking Elmo-binding activity but capable of activating G-proteins. Mechanistically, we found that expression of Myc-Elmo2 can modulate the BRET2 signal of the $G\alpha i1$-LucII/$G\beta$/$G\gamma$-GFP10 complex in cells. Second, biochemical cell fractionation and co-localization assays revealed that the Elmo-binding site of BAI3 is also required to bind Elmo and anchor it at the receptor in membrane fractionation assays. This is also consistent with the observations that direct Elmo binding to BAI1/3 is essential for fusion[20,21].

The analysis of *Bai3* knockout mice revealed defective muscle development and regeneration phenotypes that are similar to the ones reported for Bai1 mutants[21], which are mild in comparison to mutants such as *Dock1* and *Rac1*[5,6]. BAI1–3 may be redundant and compensate for one another in single knockout conditions. Dock1/Elmo/Rac signaling could also be used by multiple cell surface proteins. Generation of double/triple mutants for BAI GPCRs may be needed to fully understand their contribution to myoblast fusion. Also, a longstanding unresolved question is to determine the function of Rac1 activated by Elmo/Dock during cell fusion. Presumably, it involves reorganization of the actin cytoskeleton, but this remains to be investigated. Finally, over-expression of proteins including BAI3 and Myomaker can increase myoblast fusion such that large fibers can be obtain in culture models[12,13]. While attractive, it still remains to be demonstrated if improving the efficiency of cell fusion could improve muscle regeneration. Because GPCRs are established

pharmacological targets[53], the search for small molecules capable of activating BAI1–3 could reveal approaches to improve myoblast fusion.

## Methods

**Antibodies**. Anti-laminin DyLight 650 antibody (1:250) was obtained from Novus (Cat. NB300−144C). Mouse monoclonal anti-Myc (9E10) (1:1000) and anti-HA (1:2500) were obtained from Santa Cruz Biotechnologies (Cat. Sc-40 and SC-805-G). Mouse monoclonal anti-MyoD (1:500) was from BD Biosciences (Cat. 554130). Mouse monoclonal antibodies anti-Flag (M2) (staining 1:500, WB 1:10,000), anti-Troponin-T (1:500), anti-Desmin (1:500), anti-V5 (staining 1:250, WB 1:1000), and anti-Tubulin (1:10,000) were from Sigma-Aldrich (Cat. F1804, T6277, D1033, T5168). Rabbit polyclonal antibody anti-V5 (1:1000) was from Cell signaling (Cat. 13202). Sheep polyclonal anti-GFP antibody (1:2000) was from AbD SeroTec/Bio-Rad (Cat. 4745−1051). Monoclonal antibodies anti-Pax7 (1:10), anti-Myogenin (1:500), and anti-Myosin Heavy Chain (MF20) (staining 1:20, WB 1:10) were obtained from the Developmental Studies Hybridoma Bank (Iowa City, IA) (Cat. Pax7, F5D, MF20). Anti-Stabilin-2 (1:50) rabbit polyclonal antibody was used as previously described[19].

**Plasmids**. All plasmids, and their derivatives, are listed in Supplemental Table 1. Briefly, pSIREN-RetroQ-ZsGreen retroviral vector was obtained from Clontech. Full length human Flag-BAI3-mVenus in the pCAGG plasmid was previously reported and is a kind gift of Dr. T.C. Südhof (Stanford University)[29]. pCAGGS-Flag-BAI3-mVenus mutants (R836A, L837A, and S838G) were generated by site-directed mutagenesis (Clontech). The introduction of the N-linked glycosylation sites in the gC1q domain of C1qL4 (Q191N, N192S, Y193T, T224N, K226S) was done by mutagenesis to generate the pDisplay-HA-C1qL4GW. To generate a Glutathione S-transferase (GST)-fusion protein of C1qL4, the corresponding coding sequences were amplified by PCR from pDisplay-HA-C1qL4 (kind gift of Dr. T. Südhof (Stanford University)) and cloned into BamHI/XhoI sites of the pGEX4T1 plasmid (GE Healthcare Life Sciences). The pcDNA5-V5-Stabilin-2 plasmid was a kind gift from Dr. Harris[36] and pcDNA3-HA-C1qL1, 2, and 3 plasmids were a kind gift from Dr. Wong were previously described[27]. All additional mutants in the pCAGGS and pIRES vectors (see Supplementary Table 1 for plasmids list) were generated with the HiFi assembly kit according to

manufacturer's protocol (NEB). Insert were generated by PCR (see Supplementary table 2 for nucleotide sequence) and mixed with the vector at a ratio of 1:2. The reaction was incubated in a thermocycler at 50 °C for 15 min. Subsequently, the samples were transformed, and clones were screened and validated by sequencing.

**Stable shRNAs expression in C2C12 cells.** shRNA plasmids were constructed using the pSIREN-RetroQ-ZsGreen retroviral vector (Clontech) as previously described[20]. Several shRNA DNA oligonucleotides were designed to specifically knockdown BAI3, C1qL4, and Stabilin-2, according to the protocol accompanying the pSIREN-RetroQ retroviral vectors (see Supplementary Table 3 for nucleotide sequences). Primers were annealed and cloned into the *Bam*HI/*Eco*RI digested pSIREN-RetroQ-ZsGFP vector. Ecotropic Phoenix retrovirus packaging cells were transfected with purified pSIREN-RetroQ-ZsGreen constructs described above. The supernatants were collected 48 h post-transfection, filtered, and mixed with polybrene (Sigma) at a 5 µg/mL final concentration. C2C12 plated at low confluence were infected twice every 24 h with the viral supernatants supplemented with FBS (20% final). At $t = 48$ h, myoblasts expressing ZsGreen were sorted using a MOFLO cell sorter (Beckman Coulter). Gene knockdown efficiency was assessed by real-time Q-RT-PCR (see Supplementary Table 2 for primers used).

**Cell culture and transfections.** C2C12 (ATCC) and Sol8 (kind gift of Dr. Jacques. Drouin, IRCM) mouse myoblasts were grown in Dulbecco's Modified Eagle Medium (DMEM) supplemented with 20% FBS (vol/vol) and a mixture of penicillin and streptomycin (Gibco). C2C12 were typically transfected in six well plates (50% confluency; 4 µg of plasmid DNA) using Lipofectamine 2000 according to the manufacturer's protocol (Invitrogen). COS7, HEK293T cells (ATCC) and the retroviral packaging cell line ecoPhoenix (kind gift of Dr. André Veillette) were grown in DMEM (supplemented with 10% FBS and penicillin and streptomycin) and transfected in 10 cm tissue culture plates (50% confluence; 10 µg of plasmid DNA) using a standard calcium phosphate precipitation protocol.

**Conditioned media production.** Plasmids coding for Flag-BAI3$^{ECD}$, V5-BAI3$^{CUB}$, V5-BAI3$^{TSR}$, V5-BAI3$^{CUB+TSR}$, V5-BAI3$^{ECD}$, or HA-C1qL4 were transfected and expressed in HEK293T cells grown to confluence in a serum-free media (DMEM with penicillin and streptomycin). 48 h after transfection, supernatants were harvested and centrifuged 5 min at 201×*g*. The pellets were discarded and 10% of the conditioned media (containing the secreted protein fragments) were diluted in serum-free media and then supplemented with 2% horse serum and used in C2C12 differentiation assays.

**Mass spectrometry sample preparation.** HEK293T cells were transfected with plasmids coding for Fc-alone (control) or Fc-Flag-BAI3$^{ECD}$. 48 h following the transfection, media containing the secreted proteins were collected and filtered using a 0.5 µm filter to remove insoluble debris. Protein A Dynabeads were added to the media with gentle rotation overnight at 4 °C. To ensure that non-specific binding proteins are removed, the beads were next subjected to three washing steps, each consisting of three washes with the following buffers: Step #1 and #3: 20 mM Tris pH 7.5, 150 mM NaCl; Step #2: 20 mM Tris pH7.5, 0.5 M NaCl). The purified Fc-alone or Fc-Flag-BAI3 were then ready to use for affinity purification: these Dynabeads-coupled proteins were incubated (with rotation) with the pooled supernatant of differentiating C2C12 cells (0, 24, or 48 h) for 4 h at 4 °C. Two steps of three washes were performed (Step #1: 20 mM Tris pH 7.5, 150 mM NaCl, Step #2: 50 mM ammonium bicarbonate). Following the last wash, samples were resuspended in 50 mM ammonium bicarbonate and sent to the IRCM mass spectrometry platform for on-beads trypsin digest followed by protein identification by mass spectrometry. The Prohits suite was used to analyze the proteomics data.

**Semi-quantitative and real-time quantitative PCR (Q-RT-PCR).** Total RNA was extracted from the indicated Sol8, C2C12, and C2C12-shRNA lines as previously described[20]. Briefly, TRIZOL reagent (Invitrogen) was used according to the manufacturer protocol. Total RNAs were treated with DNAse1 (Invitrogen) and cDNAs were generated using the M-MuLV Reverse Transcriptase and random primers (NEB), as recommended by the manufacturer. Specific knockdown of the genes of interest was confirmed by real-time Q-PCR in an Mx3005P (Stratagene) system using SYBR Green PCR Master Mix (Applied Biosystems). Reaction specificity was assessed by melt curve analyses for each primer set. The TATA box gene was used as an internal control. All real-time Q-PCR reactions were carried out as follow: 5 min at 45 °C, 3 min at 95 °C, followed by 40 cycles of 15 s at 95 °C and 30 s at 60 °C. Primers used in Q-PCR analyses are described in Supplementary Table 2. C1qL family semi-quantitative PCR was performed using primers listed in the Supplementary Table 2. The PCR cycles were carried as follow: 30 s at 95 °C, followed by 30 cycles of 15 s at 95 °C and 30 s at 57°C, final extension at 68 °C for 5 min.

**Protein expression and co-immunoprecipitation (western blot).** For co-immunoprecipitation assays, HEK293T or C2C12 cells expressing the indicated proteins were lysed for 10 min in Nonidet-P40 buffer (150 mM NaCl, 50 mM Tris

pH 7.5, 1% Nonidet- P40). 500 µg of total protein extract of the indicated conditions were incubated with the indicated antibodies and Protein-A agarose and after 90 min of incubation, the beads were washed three times with lysis buffer and proteins were next eluted and analyzed by SDS-PAGE followed by immunoblotting with the indicated antibodies[20] For analysis of total cell extracts, proteins were extracted in Radio Immuno-Precipitation Assay (RIPA) buffer (50 mM Tris pH 7.5, 0,1% SDS, 0.5% deoxycholic acid, 1% Nonidet P-40 buffer, 150 mM NaCl, 5 mM EDTA) and analyzed as above. All original western blot data are presented in Supplementary Fig. 11).

**Purification of recombinant C1q domain of C1qL4.** The C1q domain of C1qL4 was expressed as a GST-fusion protein in BL21 bacteria and purified with glutathione-sepharose 4B following manufacturer recommendations (Amersham, Piscataway, NJ). The GST moiety was removed by thrombin digestion using the manufacturer digestion buffer at 4 °C for an overnight reaction (VWR). Finally, the biotinylated thrombin was removed using streptavidin beads for 30 min at 4 °C. The purified C1q domain of C1qL4 was analyzed by SDS-PAGE followed by Coomassie staining to assure protein integrity and to allow quantification. The recombinant C1qL4 protein was diluted to 100 ng/mL in C2C12 differentiation media.

**Membrane fractionation assays.** Membrane fractionation experiments were conducted as previously described[54]. Briefly, HEK293T cells were transfected with the indicated vectors (Flag-BAI3$^{WT}$, Flag-BAI3$^{\Delta N}$, Flag-BAI3$^{\Delta N/\Delta ELMO}$, V5-Stabilin-2, and Myc-ELMO2). 48 h post-transfection, cells were harvested in PBS and centrifuged at 400×*g* for 5 min. Cells pellets were suspended in Buffer A (10 mM HEPES, 15 mM MgCl₂, 10 mM KCl, 0.5 DTT and 0.05% NP-40) and a protease inhibitor cocktail. To lyse the cells, they were subjected to freeze and thaw cycles. The lysates were centrifuged, and the supernatant was kept as the cytosolic fraction. The pellets corresponding to the membrane fractions were further washed in Buffer A before extraction of the membrane proteins with 1% Triton (membrane fraction). Equal amounts of proteins of the cytosolic and membrane fractions were analyzed by Western blot.

**Chick embryos and in ovo electroporation.** Experiments using chick embryos were carried out as previously described[20] and were authorized by the Animal Care Committee of the Institut de Recherches Cliniques de Montréal in compliance with the Canadian Council of Animal Care guidelines. Briefly, fertilized chick eggs were incubated at 38.5 °C under 95% humidity for developmental staging according to standard protocols[55]. Chick somite electroporation was performed using through a small eggshell window under a Zeiss Discovery V12 stereomicroscope at stage E2.5 (between HH stage 18–19). DNA plasmids (pEGFP-N2, pCAGG-HA-C1qL4$^{WT}$ and pCAGG- HA-C1qL4$^{N-Glyc}$) were suspended in TE buffer pH 7.5 (10 mM Tris-HCl and 1 mM EDTA) at 5 µg/µL and microinjected into the somitocoele of interlimb somites (I to IV according to Scaal et al.[35]). Chick embryos were then electroporated with platinum/iridium electrodes and a TSS20 Ovodyne electroporator (settings: 25 Volts, 5 pulses) and 200 µL of a penicillin-streptomycin solution was added at the microinjection site. Shell windows were sealed with parafilm and eggs were incubated for 72 h at 38.5 °C. Embryos were then harvested and those expressing GFP were selected for further analyses.

**Muscle regeneration—cardiotoxin (CTX) injury.** Mice were anesthetized by isoflurane inhalation. 50µl of cardiotoxin from *Naja pallida* (Latoxan) (stock concentration: 10 µM) was injected in the tibialis anterior (TA) muscle of the WT and BAI3 KO mice. The TA muscles were collected 14 days following the induced-injury and processed for analysis by histology as described below.

**Histology and myofibers analysis.** TA muscles were dissected from WT (control mice) and BAI3 KO mice, and fix with 10% formalin. Muscles were embedded into paraffin blocs according to standard procedures and 5 µm sections were obtained. Hematoxylin and eosin (H&E) staining was performed and pictures were captured using a Zeiss Axiophot microscope. The cross-sectional area (CSA) of myofibers was quantified using the Volocity software (PerkinElmer Life and Analytical Sciences).

**Immunofluorescence on muscle sections.** TA muscles from WT and BAI3 KO mice were dissected and embedded with OCT compound. To analyze the number of myonuclei per fibers, 10 µm frozen sections were fixed in 4% paraformaldehyd and permeabilized with a PBS/0.2% Triton X-100 solution for 10 min. Sections were incubated in blocking buffer (PBS/1% BSA) for 1 h. Cells were next incubated with anti-laminin DyLight 650 (dilution 1:250; Novus) and Hoechst (dilution 1:10,000; Invitrogen).

To analyze the number of Pax7-positive cells, 10 µm frozen sections were fixed in 4% paraformaldehyde for 10 min. Slides were boiled for 20 min in antigen retrieval buffer (10 mM Sodium Citrate pH 6.0) prior to incubation in blocking buffer (10% Goat serum/0.4% TritonX/PBS) for 1 h. Muscle sections were incubated with primary antibody Pax7 (dilution 1:10 (Developmental Hybridoma)) diluted in 0.04% TritonX/1%BSA/PBS at 4 °C overnight. Following three washes of

PBS, sections were incubated with secondary antibody Alexa 568 conjugated goat anti-mouse secondary antibody (dilution 1:300; Invitrogen) and anti-laminin DyLight 650 (dilution 1:250; Novus) for 1 h. Hoechst (dilution 1:10,000; Invitrogen) was used to reveal nuclei. Pictures were taken with the DM6 microscope (Leica) at an objective of ×20 and the images were analyzed using the Volocity software.

**Animal experiments**. Mice used were previously described: BAI3 KO mice were generated by crossing BAI3[flox] mice[30] with *telencephalin*-Cre transgenic mice[56]. Mice were housed in a specific pathogen-free (SPF) facility and experiments were authorized by the Animal Resource Committee of Keio University. C57BL/6 mice (from The Jackson Laboratory) used for primary myoblasts isolation were housed in a specific pathogen-free (SPF) facility and experiments were authorized by the Animal Care Committee of the Institut de Recherches Cliniques de Montréal and complied with the Canadian Council of Animal Care guidelines.

**Primary myoblasts isolation**. Primary myoblasts were obtained from leg muscles of adult WT mice. Briefly, muscles were minced mechanically and digested with enzymes (trypsin and collagenase D in F12 media) at 37 °C with agitation for 1 h. Myoblasts were isolated using magnetic beads (MACS Satellite Cell Isolation Kit, together with anti-Integrin α-7 MicroBeads, Miltenyl Biotec). Primary cells were cultured on gelatin-coated dishes, in 39% DMEM with glutamax, 39% F12 with glutamax, 20% fetal bovin serum (Wisent) and 2% UltroserG (Pall Life Sciences) media. To induce myoblast differentiation, media was change for 2% horse serum in DMEM/F12 media for 72 h.

**In situ hybridization**. Chick embryos were fixed in a 4% solution of paraformaldehyde (Sigma) in PBS, equilibrated with 30% sucrose in PBS, embedded in O.C.T. (Sakura Finetek), and stored at −80 °C. Twelve micrometer sections were collected using a Leica cryostat microtome. In situ mRNA hybridization and detection were performed as described[57]. In situ mRNA hybridization images were taking using OsteoMeasure on a Leica DM 4000 light microscope at objective ×10.

**Immunofluorescence and immunohistochemistry**. C2C12 and Sol8 myoblasts were fixed in 4% paraformaldehyde and permeabilized with a PBS/0.2% Triton X-100 solution for 10 min prior to incubation in blocking buffer (PBS/1% BSA) for 1 h. Cells were next incubated with the primary antibody recognizing MyHC (MF20, dilution 1:20 (Developmental Hybridoma)) diluted in blocking buffer. Cells were washed with PBS and incubated with an Alexa 568 conjugated goat anti-mouse secondary antibody (dilution 1:2500; Invitrogen) for 1 h. Hoechst (dilution 1:10,000; Invitrogen) was used to reveal nuclei. GFP/mVenus positive chick embryos were fixed in 4% paraformaldehyde for 1 h at room temperature and washed three times with PBS. They were then incubated overnight in 30% sucrose solution at 4 °C and embedded in OCT prior to cryosectioning. 8 μm sections were stained as described above for cells with the following modifications: sections were washed in PBS/0.1% Triton X-100 and the Alexa 568 conjugated goat anti-mouse secondary antibody was used at a 1:2500 dilution. Pictures were taken with the Axiovert microscope (Zeiss) at an objective of ×20 and with a confocal microscope LSM700 (Zeiss) at an objective of × 100. In both cases, the images were analyzed using the Volocity software, as described above.

**Proximity ligation assay (PLA)**. C2C12 transfected with Flag-BAI3 and differentiated for 24 h were fixed in 4% paraformaldehyde and incubated with blocking solution (PBS/1%BSA) for one hour. Cells were next incubated with the primary antibody recognizing Flag-tag (Flag-BAI3) (M2; Sigma) and Stabilin-2[19] diluted in blocking buffer and incubated overnight at 4 °C. PLA was performed according to manufacturer protocol (Duolink PLA technology; Sigma-Aldrich). Images were acquired with a LSM700 (Zeiss) confocal microscope using a ×100 objective. Images were analyzed using the Volocity software.

**Bioluminescence resonance energy transfer 2 (BRET2)**. To conduct the BRET2 experiments, cells were transfected with either BAI3[WT], BAI3[ΔN], Myc-ELMO or CXCR4 (positive control). To generate BRET2 signal, RLucII (energy acceptor) and GFP$_{10}$ (energy donor) were fused to G proteins alpha and beta, respectively[41]. 48 h after transfection, cells were washed and the RLucII substrate Coelentrazine 400a solution (Biotum) was added and to generate light with a maximal light peak at 400 nm. BRET2 signal was measured with a BRET2$_{480-YFP}$ filter of the Mithras LB940 multimode Microplate reader (Berthold Technologies). BRET2 ratio is calculated as the light emitted by the acceptor over the light emitted by the donor. To calculate the net BRET2 signal, the background signal was subtracted from the RLucII alone transfection condition.

**Mixed population myoblast assay**. C2C12 expressing the indicated shRNAs or plasmids were stained with lipophilic cell tracking dyes PHK26 (red, pseudo-colored purple) or PHK67 (green) (Sigma-Aldrich). Staining was conducted according to the manufacturer's protocol. Cell were mixed at a ratio 1:1 and plated in 6-well plates. Differentiation was induced the next day for 48 h. Images were

acquired with an Axiovert microscope (Zeiss) at an objective of ×20. The images were analyzed using the Volocity software.

**In vitro binding assay**. HEK293T cells were transfected with HA-C1qL4 to produce conditioned media as described above. C2C12 cells expressing either Flag-BAI3 or Flag-BAI3[ΔCUB] were incubated with the HA-C1qL4-conditioned media for 10 min. The cells were then washed with PBS and fixed with 4% PFA. The cells were not permeabilized and immunostained with anti-Flag and anti-HA. Images were acquired with a LSM700 (Zeiss) confocal microscope using a × 100 objective. Images were analyzed using the Volocity software.

**Statistical analyses**. Data are expressed as mean ± standard deviation from at least three independent experiments. Statistical comparisons between samples were done with one-way ANOVA test followed by a Bonferroni test using the Prism Graph software. *P*-value < 0.05 was considered as significant.

## Data availability
The raw proteomics data, which are presented in Fig. 5 and Supplemental Fig. 6, have been uploaded to the MassIVE archive: accession number MSV000082839, or: [https://massive.ucsd.edu/ProteoSAFe/dataset.jsp?task= 184519eecd6b40a581cc0889c356bcf5]. All data that support the findings of this study are available from the corresponding author upon request.

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

## Acknowledgements

We thank Dr. D.R. Hipfner (IRCM) for critical reading of the manuscript. We recognize the technical support of Meirong Liang (IRCM) and expertise of the IRCM Microscopy (Dr. D. Filion) and Mass Spectrometry (Dr. D. Faubert) platforms. We thank Dr. Arhamatoulaye Maiga (IRIC) for guidance in BRET2 experiments. We acknowledge Dr. E.N. Harris (University of Nebraska) for the generous gift of Stabilin-2 plasmid. We acknowledge Dr. T.C. Südhof (Stanford University) for support and sharing reagents. This work was funded by an NIH grant (DK084171) to G.W.W and CIHR grants to J.-F.C. (PJT-153065), M.B. (FDN-148431) and A.K. (MOP-77556 and MOP-97758). N.H. and V.T. are recipients of Ph.D. studentships from the Fonds de Recherche du Québec-Santé (FRQ-S). M.B. holds a Canada Research Chairs in Signal Transduction and Molecular Pharmacology. J.-F.C. is a recipient of a Senior Investigator Award from the FRQ-S and holds the TRANSAT chair in Breast Cancer Research.

## Author contributions

J.-F.C., N.H. and V.T. designed the research; N.H., V.T., T.A., W.K, S.H., M.P.T. and A.P. performed the research; G.W.W., I.S.K. and M.Y. provided unique reagents; N.H., V.T., S.L., M.B., A.K. and J.-F.C. analyzed the data; J-F.C., N.H. and V.T. wrote the paper with input from all other authors.

## Additional information

**Competing interests:** The authors declare no competing interests.

