## [Peer Review File · Nature Communications]

Reviewers' Comments:

Reviewer #1:

Remarks to the Author:

In this manuscript, the authors have identified a new molecular interaction that may be involved in regulating myoblast fusion in mammalian cells. Authors have previously reported that cell surface protein BAI3 regulates myoblast fusion Elmo/Dock1 complex and Rac1. Using C2C12 myoblasts as a model system, authors demonstrate that C1qL4 protein interacts with BAI3 to inhibit myoblast fusion. Moreover, through a proteomics approach, they identified Stablin-2 as binding partner which augments GPCR activity of BAI3 to recruit Elmo proteins to augment myoblast fusion. While the role of BAI3, Stablin-2, and Elmo proteins in myoblast fusion is known, present study identifies the molecular interactions by which BAI3 promotes myoblast fusion/myogenesis. The investigators have generated several constructs which are appropriate to identify the interacting domains and partners. The manuscript is well-written and easy to follow. A major weakness in the manuscript is that many of the proteins studied here are already known to impact myoblast fusion or myogenesis. While the authors investigated molecular interactions, they are mostly in non-physiological conditions and using cell lines. Specific issues in the manuscript are as follow.

(1) Almost all the experiments are performed using C2C12 myoblasts. There are many reports that the findings using this cell line could not be reproduced in vivo or using primary myoblasts. Authors should use mouse primary myoblasts at least for some critical experiments.

(2) Similarly, there is a concern that all studies employed shRNA for knockdown. Authors should consider using primary myoblasts from knockout mice.

(3) All protein-protein interactions are studied through the overexpression of WT or mutant constructs. Authors should make an attempt to study protein-protein interaction of endogenous proteins.

(4) While they have mentioned Myomaker and Myomixer in their study, there is no experiment to determine whether these molecules are regulated through BAI3-mediated signaling during myoblast fusion. Recent studies have also identified some profusion signaling pathways in mammalian cells. Authors should investigate whether some of these profusion molecules/pathways are regulated via Stablin-2/BAI3/Elmo axis.

(5) Figure 1a uses semi-quantitative PCR to measure the mRNA levels of C1qL1-4 which is not the best approach. Authors should perform QRT-PCR or Western blot/ELISA to confirm their conclusions.

(6) Authors claim that knockdown of C1qL4 does not affect myogenesis. However, the blots presented in Supplemental figure 1g show that there is delay in the expression of MyHC and Myogenin upon knockdown of C1qL4. The blots MyoD and Myogenin look identical except that exposure time may be different. Please provide representative better quality immunoblots. Authors should also compare mRNA levels of various MRFs in control and depleted cells.

(7) Authors claim that exogenous C1qL4 inhibits myoblast fusion. However, all representative images suggest that there is inhibition in myogenesis because overall number of myotubes are reduced and there is no evidence of presence of mononucleated or bi-nucleated myoblasts (Figures 1 and 2). They should measure the levels of muscle differentiation markers. It is important to clarify whether C1qL4 inhibits myogenic differentiation or impairs only myoblast fusion. If the effect of C1qL4 is on myogenesis, then it should be clearly stated.

(8) Experiments in Figure 5 and 6 are important to understand the activation of canonical GPCR activity and membrane translocation of Elmo. However, no correlation has been made with myoblast fusion or myogenesis. Some of these experiments involving overexpressing different mutants require parallel experiments to study their effects on myoblast fusion.

Reviewer #2:

Remarks to the Author:

BAI3 is a pro-fusion molecule that induces actin nucleation through Elmo-Dock. One question about this pathway is how it is regulated to govern myoblast fusion given that BAI3 is expressed at similar levels in proliferating (fusion-incompetent) and differentiating (fusion-competent) myoblasts. The authors show that secreted C1qL4 proteins, which are known ligands for BAI3, block BAI3 activity and myoblast fusion in proliferating myoblasts. They further demonstrate that down-regulation of C1qL4 is required for fusion at least in vitro. They then use a proteomics approach to identify activating BAI3 ligands and identify stab2, recently reported to be a receptor for phosphatidylserine during myoblast fusion. Overall, the experiments are highly mechanistic as to how BAI3 coordinates myoblast fusion through ELMO2 and therefore would be interesting for the community. There are a few issues described below that, if clarified, would strengthen the manuscript.

1. One premise of the paper is that the presence of C1qL4 blocks BAI3 activity in undifferentiated myoblasts, which is supported by the idea that C1qL4 is down-regulated upon differentiation in C2C12 myoblasts (Fig. 1a) and in the chicken embryo (Fig. 2j). This analysis lacks quantification, which is needed to fully verify the results, and it is not obvious that C1qL4 is absent at day 7 of chicken muscle development. qPCR should be doable in C2C12 cells, although I'm not sure if this is true for the chicken embryo and I recognize that quantification of the in situ results in could be difficult. Perhaps higher resolution images would help or if in situ could be done for both MyoD and C1qL4 on the same section. Another possibility is to evaluate expression during mouse development, where the tissues can be easily evaluated by qPCR. Related to this, are the other C1q isoforms also expressed and regulated in vivo during muscle development?
2. Loss of C1qL4 increases the efficiency of myoblast fusion in C2C12 myoblasts, although it is not so dramatic after 48 hours and there is no effect at 24h. Could the authors comment on potential reasons for these different results? Does the fusion in cultures treated with C1qL4 shRNA eventually equal shGFP cultures at 72h?
3. The use of rC1qL4 is a powerful experiment, although there are experimental details missing that make it difficult to interpret. First, it is not clear what protein was actually used. GST-C1qL4 or cleaved C1qL4? I'm also confused as to what was used as a control. In the text it states GST but the figures say Parental. Also, the gel on supplementary figure 1j shows two bands in the GST-C1qL4 sample and what the red arrow is pointing to is not mentioned. Were the cultures treated with rC1qL4 at the initiation of differentiation and when were they assayed? What concentration of protein was used and was the effect dose-responsive?
4. One perceived weakness of the paper could be the lack of in vivo relevance. It is noted that mechanistic fusion experiments are difficult, if not impossible, in an in vivo setting. However, it would be helpful to provide more support for the idea that C1qL4 plays a functional role in vivo. It looks like there could be a differentiation phenotype, but this was not evaluated. Moreover, the timing of electroporation and sacrifice is not explained, and I am unable to unambiguously identify a multi-nucleated myofiber. Since myofiber length could be impacted by many physiological processes it would be better to quantify an endpoint most closely related to fusion. Was the electroporation done at a time when endogenous C1qL4 is absent? Is it possible to do a loss-of-function experiment for C1qL4 in the chicken embryo?
5. The identification of stabilin-2 as an activator of BAI3 is interesting, however genetic loss of stabilin-2 has a minor phenotype during mouse development. In contrast, I would expect BAI3 to have a major phenotype in the mouse (although I don't think this has been done). Does this suggest that stabilin-2 is not the most significant physiological activator of BAI3?

6. That overexpression of BAI3 rescues fusion in shStab2 treated myoblasts (Fig. 3m) does not fit in the model proposed that Stab2 activates BAI3 through a physical interaction. If Stab2 isn't present, how does BAI3 get activated? Does this suggest that Stab2 is not the most important activating factor for BAI3 as explained in comment 5?

Minor:

1. It is mentioned in the introduction and the first section of the results that both BAI1 and BAI3 levels do not change during differentiation. I'm not sure this is accurate considering the published data (Hochreiter-Hufford et al. Nature 2013) showing an increase of BAI1 at later stages of differentiation. I also don't think it's necessary to mention BAI1 in this part of the results section since only BAI3 is the subject of the experiments.

2. Supplementary Figure 1a is confusing. It shows a second C1qL4 shRNA and I'm guessing it is assayed in triplicate but this isn't clear. However there isn't a reduction of mRNA in #1 and #2, therefore I don't understand the purpose of showing those replicates.

3. Related to the above criticism that there is a lack of experimental information, it is not clear what CTL is for the conditioned media experiment in Sup Fig. 1m. Is CTL conditioned media from 293s that were transfected with an empty plasmid? I'm assuming this is the case since Fig. 2b is labeled with empty vector. Nonetheless this should be clear in the manuscript.

4. The approach to identify BAI3 interacting proteins from C2C12 cells is confusing. In the figure, it says 'supernatant' was used but in the text it states conditioned media was used.

Reviewer #3:

Remarks to the Author:

The paper "Spatiotemporal regulation of G-Protein Coupled Receptor BAI3 canonical and noncanonical signaling by C1q-Like proteins and Stabilin-2 controls myoblast fusion" [by Hamoud N, Pelletier A, Kania A, Bouvier M and Côté JF] describes new and relevant data for the research field of myoblast fusion and even to a wider field of interest including the molecular regulation of cell adhesion, cell fusion and cytoskeletal reorganization. The authors describe two proteins, the secreted C1q-like1-4 proteins and Stabilin-2, that spatiotemporally control the activity of the cell surface protein BAI3 during vertebrate myoblast fusion. The key findings of their study are that: (i) Stabilin-2 was identified as a novel BAI3-interactor; (ii) Stabilin-2/BAI3 complex functions in cis to promote myoblast fusion and that both fusing myoblasts require both receptors on their surface; (iii) BAI3 displays canonical GPCR activity; and (iv) activated G-proteins contribute to the recruitment of Elmo proteins to the membrane. The methodology of the study is described in detail allowing the work to be reproduced. Statistical analysis is appropriated. Although this work is potentially interesting, there are important points that need to be addressed to strengthen the conclusions drawn.

1 – Figure 1a – C1qL4 are secreted proteins and therefore expression levels of mRNA by semi-quantitative RT-PCR are not sufficient to prove that these proteins are in fact been translated and secreted in muscle cells (C2C12 and/or in Sol8 myogenic cultures). There are commercially available antibodies against C1qL4 proteins which could be used in Western blots and/or immunofluorescence staining of muscle cells. Therefore, authors cannot claim in the description of Figure 7 (Discussion) that "proliferating and early differentiating myoblasts express C1qL4 to maintain BAI3 inactive", unless they show that C1qL4 proteins are indeed expressed in proliferating myoblasts and not in myotubes.

2 – Figure 1b:

(i) Myosin heavy chain (MyHC) is a myofibrillar protein and any immunofluorescence labeling of multinucleated myotubes, such as the ones shown in Figure 1b (magnification), is expected to

display the characteristic sarcomeric A-band staining. Figure 1b shows only a haze labeling of MyHC in C2C12 muscle fibers which implies that these myotubes are not organizing normal sarcomeres.

(ii) It looks like the depletion of C1qL4 leads to an increase in muscle fiber size in C2C12 cell cultures (Figure 1b), which suggest muscle hypertrophy. Authors quantified only myoblast fusion index. Depletion of C1qL4 could be leading to the formation of myotubes with an abnormal ratio between cytoplasm/nucleus areas and an increase in total cell area.

(iii) the dotted white boxes shown in Figure 1b do not correspond to the exact images shown at higher magnifications.

3 – Figure 2 – the interaction between C1qL4 and BAI3 was demonstrated only by immunoprecipitation (in cell-free experiments) and therefore care should be taken when authors claim that “These results underline the importance of C1qL4 binding to BAI3 to inhibit myoblast fusion”. They did not show a co-localization of these two proteins in cells. There is also the possibility that in muscle cells the interaction between C1qL4 and BAI3 could be indirect and mediated by other unknown proteins, which was not discussed by the authors.

4 - Supplementary Figure 3c – Stabilin-2 was identified in conditioned media of differentiating C2C12 cells by mass spectrometry (Supplementary Figure 3c) and authors stated that “...Stabilin-2 found at the cell surface”. Since Stabilin-2 is a transmembrane protein how its presence in the conditioned media of C2C12 cells could be explained? Proteolytic cleavage of membrane proteins? It is also not clear whether secreted Stabilin-2 proteins could activate BAI3 receptors.

5 – Authors did not analyze the expression and distribution of BAI3 receptor proteins during the differentiation of C2C12 muscle cells. In a previous report they showed that BAI3 transcripts are expressed in C2C12 cells before and after differentiation at the same levels (Hamoud et al., 2014). There are commercially available antibodies against BAI3 proteins which could be used in Western blots and/or immunofluorescence staining of muscle cells. Immunofluorescence labeling of BAI3 proteins in cultured muscle cells could also reveal their spatial organization (in dots, patches or continuous lines) at the sarcolemma before, during and after myoblast fusion, and provide valuable information.

6 – Figure 4 – The mixed population myoblast assay is an elegant way to show that BAI3 and Stabilin-2 interact in cis in myoblasts to promote cell fusion. Nevertheless, authors analyzed only myoblast-myoblast fusion and they did not mention myoblast fusion with pre-existing multinucleated myotubes. It has been well described by different groups that myoblast-myoblast fusion is not quite the same as myoblast-myotube fusion. Do BAI3 and Stabilin-2 interact in the same way in myoblast-myotube adhesion to promote fusion? Do myotubes have both BAI3 and Stabilin-2 in their plasma membrane? Antibodies against BAI3 and Stabilin-2 combined with immunofluorescence could help to answer these questions.

7 – Figure 5 and 6 – All the experiments shown in Figure 5 and 6 were performed with non-muscle cells (HEK293T cells) and therefore there is no confirmation that muscle cells (C2C12 or Sol8 cells) will behave in the same way. So, all data showing that beta-arrestin co-localizes (or not) with BAI3 at the plasma membrane and that Stabilin-2 promotes the GPCR activity of BAI3 needs to be further demonstrated in muscle cells.

8 – Figure 5i – Figure 5i data do not support authors claim that “These experiments revealed that BAI3(Δ N), similar to RhoG, efficiently promoted the translocation of Myc-Elmo2 in the membrane biochemical fraction”. Figure 5i does not show that BAI3(Δ N) efficiently promoted the translocation of Myc-Elmo2 in the membrane fraction.

9 – Figure 7:

(i) In their working model, authors describe that “Stabilin-2, bound to phosphatidylserine exposed on the target myoblast enters into a heterodimeric complex with BAI3 that promote its GPCR activity and G-protein-mediated recruitment of Elmo”, but there is no experiment in their work showing that phosphatidylserine exposure induce GPCR activity and G-protein-mediated recruitment of Elmo. These experiments could reinforce their working hypothesis.

(ii) Authors suggest that “Remodeling of the cytoskeleton by the Stabilin-2/BAI3/Elmo/Dock/Rac1 pathway could be involved to drive protrusions to further increase the proximity of the myoblast membranes, similar to what is observed in *Drosophila*”. This is an important data that could improve the present work. Analysis of membrane protrusions by confocal microscopy using

fluorescently labeled Phalloidin could confirm whether a similar role for actin cytoskeleton reorganization induced by the Stabilin-2/BAI3/Elmo/Dock/Rac1 pathway is observed in vertebrate muscle cells.

10 – Supplementary Figure 1a,b,c – shC1qL4#2-1 and shC1qL4#2-2 do not induce a downregulation of C1qL4 mRNAs (and neither a change in cell fusion index). Authors did not discuss these results.

11 - Supplementary Figure 1d,e – Sol8 is a myogenic cell line isolated from primary cultures of soleus muscle with a phenotype of slow twitch fibers. Depletion of C1qL4 in these cells increases myoblast fusion. Could C1qL4 have a major role in myoblast fusion in slow fibers but not in fast twitch fibers?

12 – Authors did not mention in the Introduction the well-established role of cadherin/beta-catenin adhesion complexes during vertebrate myoblast fusion. They only describe the recently discovered set of vertebrate cell surface proteins that control myoblast when they mention that “DOCK1, the GTPase Rac1 and the actin nucleator N-WASP have been demonstrated to play an evolutionarily conserved and essential role in cell-cell fusion in vivo in mice”. Given that cadherins are highly concentrated at membrane sites of pre-fusion myoblasts, they might even interact with BAI3/Stabilin-2 membrane domains.

REVIEWER #1

“...The manuscript is well-written and easy to follow. A major weakness in the manuscript is that many of the proteins studied here are already known to impact myoblast fusion or myogenesis. While the authors investigated molecular interactions, they are mostly in non-physiological conditions and using cell lines. Specific issues in the manuscript are as follow.”

We thank this reviewer for an excellent appraisal of our manuscript and pointing out some weaknesses. We have now addressed the majority of the comments that were raised (see below).

(1) “Almost all the experiments are performed using C2C12 myoblasts. There are many reports that the findings using this cell line could not be reproduced *in vivo* or using primary myoblasts. Authors should use mouse primary myoblasts at least for some critical experiments.”

We now include additional experiments using primary myoblasts from wild-type mice to characterize the role of C1qL4 on fusion. As shown in Figure 1 n-o, primary myoblasts treated with recombinant C1qL4 exhibited significantly less fusion than control primary myoblasts. This result confirms our initial observations with C2C12 cells demonstrating that C1qL4 negatively regulates cell fusion.

(2) “Similarly, there is a concern that all studies employed shRNA for knockdown. Authors should consider using primary myoblasts from knockout mice.”

This comment prompted us to initiate a collaboration with Dr. Michisuke Yuzaki’s laboratory (Japan). Dr. Yuzaki generously hosted one of our trainee in his laboratory and made available a significant number of control and mutant *Bai3* animals. Our study now includes detailed analyses of muscles in adult *Bai3* knock-out mice. We also challenged these mice to cardiotoxin-induced tissue injury to analyze muscle regeneration.

First, we compared TA muscles from wild-type and *Bai3* KO young adult mice. These experiments revealed that the KO mice presented significantly smaller fibers compared to the wild-type controls. Second, to gain insights into the implication of *Bai3* during muscle regeneration, we subjected the wild-type and *Bai3* KO mice to cardiotoxin-induced tissue injury and assessed muscle regeneration. We observed that the *Bai3* KO mice regenerate smaller myofibers in comparison to wild-type mice (14 days post-injury). These results provide *in vivo* evidence for a contribution of *Bai3* to myoblast fusion during both development and regeneration. These experiments are included in Figure 1 (a-f).

Unfortunately, we were limited by the number of mice that could be made available for this project by Dr. Yuzaki’s laboratory. We decided to focus on the *in vivo* experiments as they are more informative of the contribution of *Bai3* to myogenesis and regeneration. We have now exhausted the *Bai3* KO mouse colony of Dr. Yuzaki to include the experiments presented in this revised manuscript. While we requested importing these mice to our laboratory, we are several months away to be able to conduct additional experiments given the low number of animals currently available which are also needed in the Yuzaki laboratory for other research projects.

(3) “All protein-protein interactions are studied through the overexpression of WT or mutant constructs. Authors should make an attempt to study protein-protein interaction of endogenous proteins.”

The complexity of carrying out traditional biochemical analyses on GPCRs, which are even more challenging for the atypical Adhesion GPCRs due to their large size, is by now well established. These GPCRs, when solubilized, are prone to aggregation (due to the 7 hydrophobic transmembrane domains) rendering them poorly soluble and inadequately amenable to classical biochemical studies. To circumvent

these limitations, most laboratories studying GPCRs now rely on approaches using biosensors, such as BRET-based reporters that were developed by our collaborator and international leader in this area of research, Dr. Michel Bouvier. This is in fact the gold standard in the field to define GPCR coupling to hetero-trimeric G-proteins, as we did in this paper to demonstrate that BAI3 displays GPCR catalytic activity.

We still considered this point of the reviewer seriously as we are fully aware of this limitation of our study pointed out by this reviewer. Unfortunately, we have tested a large number of BAI3 antibodies and none provided a reliable signal in immunoprecipitation or western blot for endogenous BAI3. We have however obtained an anti-Stabilin-2 antibody from new collaborator Dr. In-San Kim that performs well to detect Stabilin-2 in C2C12 myoblasts (as they had reported in *Nature Communications* 2016). To assess the interaction of endogenous Stabilin-2 with BAI3, we successfully carried out a Proximity Ligation Assay (PLA) experiment that revealed that endogenous Stabilin-2 is in complex with BAI3 by transfecting a tracer amount of exogenous Flag-BAI3 in C2C12 cells (Supplementary Figure 5f).

(4) While they have mentioned Myomaker and Myomixer in their study, there is no experiment to determine whether these molecules are regulated through BAI3-mediated signaling during myoblast fusion. Recent studies have also identified some profusion signaling pathways in mammalian cells. Authors should investigate whether some of these profusion molecules/pathways are regulated via Stabilin-2/BAI3/Elmo axis.

These experiments are excellent suggestions to define whether the lack of fusion observed by manipulating BAI3 or C1qL4 levels was imputable to decreased expression of the fusogenic proteins. As requested, we investigated the expression of Myomaker and Myomixer by Q-RT-PCR following differentiation upon manipulation of the cells by depletion of C1qL4, BAI3 or Stabilin-2. These experiments revealed that C1qL4, BAI3 or Stabilin-2 depletion did not impair myoblast fusion via blocking the expression of Myomaker and Myomixer (Supplementary Figure 2e-f and Supplementary Figure 5g-h).

(5) Figure 1a uses semi-quantitative PCR to measure the mRNA levels of C1qL1-4 which is not the best approach. Authors should perform QRT-PCR or Western blot/ELISA to confirm their conclusions.

We now include a Q-RT-PCR analysis of *C1qL1-4* expression at different time points of differentiation. In agreement with the RT-PCR data, we show that *C1qL4* is the member of the family that is the most highly expressed during myoblast proliferation and that its expression decreases upon differentiation (Supplementary Figure 1b). Please also see our answer to Reviewer 3; we have not been able to identify a reliable anti-C1qL4 antibody.

(6) Authors claim that knockdown of C1qL4 does not affect myogenesis. However, the blots presented in Supplemental figure 1g show that there is delay in the expression of MyHC and Myogenin upon knockdown of C1qL4. The blots MyoD and Myogenin look identical except that exposure time may be different. Please provide representative better quality immunoblots. Authors should also compare mRNA levels of various MRFs in control and depleted cells.

We performed new western blot experiments (MyHC, MyoD and Troponin T) as well as Q-RT-PCR analyses (*MyoD*, *Myogenin*, *Myosin Heavy Chain 4*) to characterize myoblast differentiation in the following conditions: Parental, shGFP and shC1qL4 C2C12 cells. These experiments confirmed that myoblast differentiation is normal upon depletion C1qL4. The Q-RT-PCR analyses revealed that *MyHC4* expression is increased at day 2-4 of differentiation in shC1qL4 cells in comparison to the control myoblasts which is in line with more fusion (Supplementary Figure 2a-d).

(7) Authors claim that exogenous C1qL4 inhibits myoblast fusion. However, all representative images suggest that there is inhibition in myogenesis because overall number of myotubes are reduced and there is no evidence of presence of mononucleated or bi-nucleated myoblasts (Figures 1 and 2). They should measure the levels of muscle differentiation markers. It is important to clarify whether C1qL4 inhibits myogenic differentiation or impairs only myoblast fusion. If the effect of C1qL4 is on myogenesis, then it should be clearly stated.

To address this valid question, we treated C2C12 cells with C1qL4 during differentiation and analyzed the expression of differentiation markers. We report that the decrease in formation of multinucleated myotubes when C2C12 cells are exposed to C1qL4 is a result of impaired myoblast fusion since observed that differentiation appears unaffected as per assessment of expression of a panel of markers (MyoD, Myogenin, MyHC4; Supplementary Figure 3f).

(8) Experiments in Figure 5 and 6 are important to understand the activation of canonical GPCR activity and membrane translocation of Elmo. However, no correlation has been made with myoblast fusion or myogenesis. Some of these experiments involving overexpressing different mutants require parallel experiments to study their effects on myoblast fusion.

To address this important comment, we carried out some of the critical structure/functions experiments done in non-myogenic cell lines back into C2C12 myoblasts. We now demonstrate that coupling of BAI3 to Stabilin-2 is important functionally: we found that the co-expression of Stabilin-2 together with the BAI3 mutant lacking its extracellular domain (i.e. deficient in Stabilin-2-binding) decreases myoblast fusion. This data suggests that the BAI3/Stabilin-2 coupling is essential for these proteins to promote myoblast fusion. Furthermore, we tested if BAI3 requires its Elmo-binding activity to synergize with Stabilin-2 to promote cell fusion. We found that the BAI3 mutant lacking Elmo-binding activity completely failed to cooperate with Stabilin-2 to promote cell fusion. Collectively, these new results reinforce the importance of the formation of a Stabilin-2/BAI3/Elmo complex to promote myoblast fusion (Figure 6g-h).

REVIEWER #2

“BAI3 is a pro-fusion molecule that induces actin nucleation through Elmo-Dock. (...) Overall, the experiments are highly mechanistic as to how BAI3 coordinates myoblast fusion through ELMO2 and therefore would be interesting for the community. There are a few issues described below that, if clarified, would strengthen the manuscript.”

We thank this reviewer for a positive evaluation of our work and for raising constructive critics. We would however like to point out that there are currently no published work demonstrating that BAI-family GPCRs induce actin nucleation through Elmo-Dock proteins. The exact contribution of Elmo-Dock-Rac to actin dynamics remains an unexplored area of research that will be important to address in the future.

“1. One premise of the paper is that the presence of C1qL4 blocks BAI3 activity in undifferentiated myoblasts, which is supported by the idea that C1qL4 is down-regulated upon differentiation in C2C12 myoblasts (Fig. 1a) and in the chicken embryo (Fig. 2j). This analysis lacks quantification, which is needed to fully verify the results, and it is not obvious that C1qL4 is absent at day 7 of chicken muscle development. qPCR should be doable in C2C12 cells, although I'm not sure if this is true for the chicken embryo and I recognize that quantification of the in situ results in could be difficult. Perhaps higher resolution images would help or if in situs could be done for both MyoD and C1qL4 on the same section. Another possibility is to evaluate expression during mouse development, where the tissues can be easily evaluated by qPCR. Related to this, are the other C1q isoforms also expressed and regulated in vivo during muscle development?”

We now include in the revised manuscript quantitative RT-PCR analyses to characterize the expression levels of C1qL-family members during C2C12 differentiation (see our answer to Reviewer 1 (comment #5) and Supplementary Figure 1b). We are aware that the quantification of C1qL4 expression *in vivo* was a limitation of our study and we agree with the reviewer's assessment of these experiments. We carried out new *in situ* hybridization experiments (*Pax3*, *MyoD* and *C1qL4*) on serial sections of chick embryo tissues at different developmental stages (E4-E7). Our improved experiments now clearly show that *C1qL4* is expressed in the myogenic muscle through the development (Figure 2j). Within this developmental window, we now conclude that *C1qL4* is expressed and not necessarily shutdown as observed in C2C12 cells. We also further discuss this and its implications in the manuscript. Finally, we tried to generate ISH RNA probes for *C1qL1-3* despite the fact that the chicken sequences are poorly annotated. Unfortunately, none of these probes generated a specific signal in ISH. Therefore, at this time, we can only speculate on the contribution of C1qL1-3 to the regulation of myogenesis.

“2. Loss of C1qL4 increases the efficiency of myoblast fusion in C2C12 myoblasts, although it is not so dramatic after 48 hours and there is no effect at 24h. Could the authors comment on potential reasons for these different results? Does the fusion in cultures treated with C1qL4 shRNA eventually equal shGFP cultures at 72h?”

To answer this question, we conducted a differentiation assay and monitored fusion at t=0,24,36,48 and 72h. We observed that shGFP and shC1qL4 cells behaved similarly until 36 hours of differentiation. At 48 hours post-differentiation, shC1qL4-depleted cells began exhibiting a higher fusion index in comparison to shGFP control cells (as initially reported in the first version of the manuscript). This difference is still statistically significant at 72 hours post-differentiation although we can observe that shC1qL4 cells have reached a plateau of fusion while the control shGFP cells still undergo fusion between 48-72 hours. Hence, we conclude that depletion of C1qL4 increases fusion and further discuss these results in the text. This new data is presented in Supplementary Figure 2g-h.

“3. The use of rC1qL4 is a powerful experiment, although there are experimental details missing that make it difficult to interpret. First, it is not clear what protein was actually used. GST-C1qL4 or cleaved C1qL4? I'm also confused as to what was used as a control. In the text it states GST but the figures say Parental. Also, the gel on supplementary figure 1j shows two bands in the GST-C1qL4 sample and what the red arrow is pointing to is not mentioned. Were the cultures treated with rC1qL4 at the initiation of differentiation and when were they assayed? What concentration of protein was used and was the effect dose-responsive?”

Indeed, our initial description of how we prepared purified recombinant C1qL4 was not completely clear. Briefly, we followed the protocol from Dr. Sudh f's laboratory (PNAS 2011) that generously gave us the plasmids. We purified the GST-tagged C1q domain of C1qL4 produced in *e. coli* and cleaved the GST tag by incubation with thrombin (which was removed post-digestion). The gel shown in Supplementary Figure 3a demonstrates that the digestion was complete as we detect the GST moiety (approximately 26 kDa) and the C1q domain of C1qL4 (approximately 15 kDa). We have not extensively quantified the recombinant protein, but we estimated, by comparing to BSA standards) that we added approximately 100 ng per ml of media during myoblast differentiation. We also found that GST-bound C1qL4 was as efficient to block cell fusion (not shown). Finally, we also prepared conditioned media containing secreted HA-C1qL4 and its BAI3-binding deficient mutant. We conducted dose response assays in pilot experiments to identify the ideal amount of media to use for each experiment (we used 10% of conditioned media in the differentiation media; see Supplementary Figure 3b). We now clarified this point extensively in the text, legend and material and method sections.

“4. One perceived weakness of the paper could be the lack of *in vivo* relevance. It is noted that

mechanistic fusion experiments are difficult, if not impossible, in an *in vivo* setting. However, it would be helpful to provide more support for the idea that C1qL4 plays a functional role *in vivo*. It looks like there could be a differentiation phenotype, but this was not evaluated. Moreover, the timing of electroporation and sacrifice is not explained, and I am unable to unambiguously identify a multi-nucleated myofiber. Since myofiber length could be impacted by many physiological processes it would be better to quantify an endpoint most closely related to fusion. Was the electroporation done at a time when endogenous C1qL4 is absent? Is it possible to do a loss-of-function experiment for C1qL4 in the chicken embryo?"

We have now carefully described the timing of electroporation and sacrifice in the method section. Our new *in situ* hybridization experiments (see point #1 above) suggest that unlike in C2C12, the mRNA of *C1qL4* is not sharply decreasing during differentiation *in vivo* (Figure 2j). In our *in vivo* somite electroporation experiments, we concluded that expression of HA-C1qL4 led to a block in myoblast fusion, which is not observed by expressing the HA-C1qL4 deficient in BAI3-binding. As stated by this reviewer, we agree that we cannot totally rule out that HA-C1qL4 is not acting on cell differentiation rather than, or in addition to, fusion. We have stained the tissue section against both Myogenin and MyHC and found that the cells remain mono-nucleated and express both differentiation markers (Figure 2k-l and Supplementary Figure 4e-f). In addition to our analyses in C2C12 cells that demonstrated that treatment with recombinant C1qL4 does not impair differentiation (see our answer to comment #7 of Reviewer 1 and Supplementary Figure 3f), we conclude that fusion, but not differentiation, is affected in the chick embryo myoblast fusion model.

"5. The identification of stabilin-2 as an activator of BAI3 is interesting, however genetic loss of stabilin-2 has a minor phenotype during mouse development. In contrast, I would expect BAI3 to have a major phenotype in the mouse (although I don't think this has been done). Does this suggest that stabilin-2 is not the most significant physiological activator of BAI3?"

As answered to Reviewer 1 (see comment #2), we collaborated with Dr. Michisuke Yuzaki and report in the revised manuscript a significant but minor contribution of *Bai3* to muscle development and regeneration using a *Bai3* knock-out mouse model (Figure 1a-f). The phenotype that we observed is highly similar to the knockout of either *Stabilin-2* (*Nature Communications* 2016) or *Bai1* (*Nature* 2013). We suspect that it may be important to generate double or triple knockouts of *Bai*-family receptors to observe a major block of fusion similar to what was observed in *Myomaker*, *Myomixer*, *Dock1* or *Rac1* mutants. Likewise, generate double mutant mice of *Stabilin-1* and *Stabilin-2* may be needed to uncover the functional redundancy of these receptors during myoblast fusion. These experiments go beyond the scope of our current study. As such, we don't think it is possible at this time to conclude that *Stabilin-2* is not the most significant physiological activator of *Bai3*. We also do not exclude the possibility that multiple cell surface proteins could be interacting with the *Elmo/Dock* pathway, including *Myomaker/Myomixer*, *CDO/BOC* or *integrins*. Clearly, more research is needed to answer these central questions in the field.

"6. That overexpression of BAI3 rescues fusion in shStab2 treated myoblasts (Fig. 3m) does not fit in the model proposed that Stab2 activates BAI3 through a physical interaction. If Stab2 isn't present, how does BAI3 get activated? Does this suggest that Stab2 is not the most important activating factor for BAI3 as explained in comment 5?"

We have several explanations to address this point. The experiments discussed by the reviewer involve gene knockdown of *Stabilin-2* and residual mRNA and proteins are surely present. It is also possible *Stabilin-1* is present in C2C12 and compensate for the decrease of *Stabilin-2* expression. Finally, we cannot rule out that a pool of BAI3 expressed in *Stabilin-2*-depleted cells becomes activated due to overexpression. We are confident that these experiments, together with biochemical and functional studies, demonstrate that *Stabilin-2* can activate BAI3 to promote fusion.

“Minor Comment 1. It is mentioned in the introduction and the first section of the results that both BAI1 and BAI3 levels do not change during differentiation. I’m not sure this is accurate considering the published data (Hochreiter-Hufford et al. Nature 2013) showing an increase of BAI1 at later stages of differentiation. I also don’t think it’s necessary to mention BAI1 in this part of the results section since only BAI3 is the subject of the experiments.”

As suggested, we have removed the reference to BAI1 in this section.

“Minor Comment 2. Supplementary Figure 1a is confusing. It shows a second C1qL4 shRNA and I’m guessing it is assayed in triplicate but this isn’t clear. However there isn’t a reduction of mRNA in #1 and #2, therefore I don’t understand the purpose of showing those replicates.”

We had included all the tested shRNAs (including the ones that were inefficient) in our initial submission. We now clarified this in the manuscript only include the results with the additional functional shRNA for C1qL4 tested in both C2C12 and Sol8 cells to remove the confusion.

“Minor Comment 3. Related to the above criticism that there is a lack of experimental information, it is not clear what CTL is for the conditioned media experiment in Sup Fig. 1m. Is CTL conditioned media from 293s that were transfected with an empty plasmid? I’m assuming this is the case since Fig. 2b is labeled with empty vector. Nonetheless this should be clear in the manuscript.”

Yes, the assumption of the reviewer was the right one. We have now clarified this in the text.

“Minor Comment 4. The approach to identify BAI3 interacting proteins from C2C12 cells is confusing. In the figure, it says ‘supernatant’ was used but in the text it states conditioned media was used.”

We thank the reviewer for raising this point. We have corrected this.

REVIEWER #3

“The paper ... describes new and relevant data for the research field of myoblast fusion and even to a wider field of interest including the molecular regulation of cell adhesion, cell fusion and cytoskeletal reorganization. ... The methodology of the study is described in detail allowing the work to be reproduced. Statistical analysis is appropriated. Although the work is potentially interesting, there are important points that need to be addressed to strengthen the conclusions drawn.”

We thank this reviewer for the positive comments and critical evaluation of our work. Below, we answer the comments raised in the review.

“1 – Figure 1a – C1qL4 are secreted proteins and therefore expression levels of mRNA by semi-quantitative RT-PCR are not sufficient to prove that these proteins are in fact been translated and secreted in muscle cells (C2C12 and/or in Sol8 myogenic cultures). There are commercially available antibodies against C1qL4 proteins which could be used in Western blots and/or immunofluorescence staining of muscle cells. Therefore, authors cannot claim in the description of Figure 7 (Discussion) that “proliferating and early differentiating myoblasts express C1qL4 to maintain BAI3 inactive”, unless they show that C1qL4 proteins are indeed expressed in proliferating myoblasts and not in myotubes.”

We thank the reviewer for highlighting this point. We now include a Q-RT-PCR of C1qL4 at different time point of differentiation (Supplementary Figure 1b). We observed, similar to the RT-PCR data, that C1qL4 is the most expressed member during proliferation (d=0) and that its expression decreases when differentiation is initiated and stays low when myotubes are formed (Supplementary figure 1a-b).

Moreover, we tried to detect endogenous C1qL4 by Western blot or staining with different antibodies obtained commercially (Santa Cruz Biotechnology catalog #: sc-13811 and Invitrogen catalog #: PA5-68158). Unfortunately, no specific signal was obtained using these reagents. Therefore, this is a major technical limitation that prevented us to address in detail this comment from the reviewer.

“2 – Figure 1b:(i) Myosin heavy chain (MyHC) is a myofibrillar protein and any immunofluorescence labeling of multinucleated myotubes, such as the ones shown in Figure 1b (magnification), is expected to display the characteristic sarcomeric A-band staining. Figure 1b shows only a haze labeling of MyHC in C2C12 muscle fibers which implies that these myotubes are not organizing normal sarcomeres.”

In the conditions and timeframe that we differentiate our C2C12 cells, we agree with this reviewer that we do not detect a clear sarcomeric A-band organisation of MyHC. However, we have found similar MyHC staining in other publications using C2C12 cells (e.g. the Stabilin-2 (Nat. Comms 2016) and Myomerger (Nat. Comms 2017) papers). In contrast, our *in vivo* studies (Laurin M (PNAS 2008) and our unpublished work in progress), we do detect beautifully organized sarcomeric bands. We are confident in our staining against MyHC that we primarily use to assess cell fusion.

“(ii) It looks like the depletion of C1qL4 leads to an increase in muscle fiber size in C2C12 cell cultures (Figure 1b), which suggest muscle hypertrophy. Authors quantified only myoblast fusion index. Depletion of C1qL4 could be leading to the formation of myotubes with an abnormal ratio between cytoplasm/nucleus areas and an increase in total cell area.”

This is an interesting comment. We now include quantifications to address the hypertrophy hypothesis. We found that the depletion of C1qL4 leads to an increase in fibers size as determined by quantifying the total cell area (Supplementary Figure 2j). To address if this was a result of hypertrophy or an increase in myoblast fusion, we then calculated the ratio of the fibers cytosol area to nuclei area. These analyses revealed that there is no significant difference between *shC1qL4* and the control cell lines (Supplementary Figure 2i). Thus, we conclude depletion of C1qL4 increases myoblast fusion rather than promoting hypertrophy.

“(iii) the dotted white boxes shown in Figure 1b do not correspond to the exact images shown at higher magnifications.”

We thank the reviewer for pointing this out and we have now corrected the dotted white boxes to make sure they correspond to the correct images.

“3 – Figure 2 – the interaction between C1qL4 and BAI3 was demonstrated only by immunoprecipitation (in cell-free experiments) and therefore care should be taken when authors claim that “These results underline the importance of C1qL4 binding to BAI3 to inhibit myoblast fusion”. They did not show a co-localization of these two proteins in cells. There is also the possibility that in muscle cells the interaction between C1qL4 and BAI3 could be indirect and mediated by other unknown proteins, which was not discussed by the authors.”

We thank the reviewer for raising this interesting comment. We now include an experiment where we assay C1qL4 binding to C2C12 cells. We used C2C12 overexpressing either BAI3^{WT} or BAI3^{ΔCUB}. The cells were then incubated with conditioned media containing HA-C1qL4 for 10 minutes followed by

washes with PBS before assessing HA-C1qL4 binding at the cell surface (anti-HA staining). As shown in Supplementary figure 4b, we observed accumulation of HA-C1qL4 at the surface of cells expressing BAI3^{WT} but not BAI3^{ACUB}. This result, together with the immunoprecipitation data, suggests that BAI3 and C1qL4 can directly interact.

“4 - Supplementary Figure 3c – Stabilin-2 was identified in conditioned media of differentiating C2C12 cells by mass spectrometry (Supplementary Figure 3c) and authors stated that “...Stabilin-2 found at the cell surface”. Since Stabilin-2 is a transmembrane protein how its presence in the conditioned media of C2C12 cells could be explained? Proteolytic cleavage of membrane proteins? It is also not clear whether secreted Stabilin-2 proteins could activate BAI3 receptors.”

We thank the reviewer for highlighting this point (similar Minor Comment #4 of Reviewer 2). We concur that we did not clearly explain our method and the data. Indeed, we identified Stabilin-2 from the supernatant of differentiating C2C12 cells. Much like the reviewer, we were initially surprised by this observation, but we found that it was reported that the ectodomain of Stabilin-2 can be cleaved and shed (Harris HN et al. (JBC) 2007). Hence, we conclude that we identified a cleaved fragment of Stabilin-2. We have conducted a series of experiments in the paper that confirm the interaction and explain the molecular detail of the complex formation. Finally, we have not tested whether a soluble ectodomain of Stabilin-2 could activate fusion. This is an interesting comment that we will certainly pursue in the future but we felt that it was more appropriate to restrict our analyses to full length Stabilin-2 in this manuscript.

“5 – Authors did not analyze the expression and distribution of BAI3 receptor proteins during the differentiation of C2C12 muscle cells. In a previous report they showed that BAI3 transcripts are expressed in C2C12 cells before and after differentiation at the same levels (Hamoud et al., 2014). There are commercially available antibodies against BAI3 proteins which could be used in Western blots and/or immunofluorescence staining of muscle cells. Immunofluorescence labeling of BAI3 proteins in cultured muscle cells could also reveal their spatial organization (in dots, patches or continuous lines) at the sarcolemma before, during and after myoblast fusion, and provide valuable information.”

We are fully aware of this valid point raised by the reviewer. Unfortunately, we have generated our own anti-BAI3 antibody and tested a number of commercial BAI3 antibodies (Santa Cruz Biotechnology Catalog #: sc-66817, Sigma Catalog #: SAB4502524, Thermo Fisher Catalog #: PA5-67719) and none provided a reliable signal on the endogenous protein. Therefore, the protein-protein interactions were largely studied in overexpression conditions. We have now included a new experiment where we can detect “tracer” Flag BAI3 interacting with endogenous Stabilin-2 using the Proximity Ligation Assay (PLA) technology (see our answer to a similar Comment (#3) from Reviewer 1). This new data is presented in Supplementary Figure 5f.

“6 – Figure 4 – The mixed population myoblast assay is an elegant way to show that BAI3 and Stabilin-2 interact in cis in myoblasts to promote cell fusion. Nevertheless, authors analyzed only myoblast-myoblast fusion and they did not mention myoblast fusion with pre-existing multinucleated myotubes. It has been well described by different groups that myoblast-myoblast fusion is not quite the same as myoblast-myotube fusion. Do BAI3 and Stabilin-2 interact in the same way in myoblast-myotube adhesion to promote fusion? Do myotubes have both BAI3 and Stabilin-2 in their plasma membrane? Antibodies against BAI3 and Stabilin-2 combined with immunofluorescence could help to answer these questions.”

This is an interesting and valid point. To address this, we conducted mix populations experiments with pre-formed myotubes. Single myoblasts were then added and fusion with the pre-formed myotubes was assessed. We observed that both receptors are required on both fusing partners; see Supplementary Figure

6a-g. These results highlight that BAI3 and Stabilin-2 are important for myoblast-myoblast and for myoblast-myotube fusion.

“7 – Figure 5 and 6 – All the experiments shown in Figure 5 and 6 were performed with non-muscle cells (HEK293T cells) and therefore there is no confirmation that muscle cells (C2C12 or Sol8 cells) will behave in the same way. So, all data showing that beta-arrestin co-localizes (or not) with BAI3 at the plasma membrane and that Stabilin-2 promotes the GPCR activity of BAI3 needs to be further demonstrated in muscle cells.”

All the immunofluorescence showing the recruitment of β -arrestin by the active form of BAI3, the recruitment of ELMO2 by BAI3 active and finally the promotion of the GPCR activity of BAI3 by Stabilin-2 were repeated in C2C12 and now included in the manuscript (Figure 5b,e-h and 6b-e). These experiments further support our initial conclusions.

“8 – Figure 5i – Figure 5i data do not support authors claim that “These experiments revealed that BAI3(Δ N), similar to RhoG, efficiently promoted the translocation of Myc-Elmo2 in the membrane biochemical fraction”. Figure 5i does not show that BAI3(Δ N) efficiently promoted the translocation of Myc-Elmo2 in the membrane fraction.”

We agree with this reviewer that we exaggerated our interpretation of the data by saying “efficiently promoted”. The recruitment observed is reproducible, but we appreciate that it is modest when compared to the recruitment of Elmo at the membrane when we co-express BAI3 and Stabilin-2. We corrected the text to accurately describe these observations.

“9 – Figure 7:

(i) In their working model, authors describe that “Stabilin-2, bound to phosphatidylserine exposed on the target myoblast enters into a heterodimeric complex with BAI3 that promote its GPCR activity and G-protein-mediated recruitment of Elmo”, but there is no experiment in their work showing that phosphatidylserine exposure induce GPCR activity and G-protein-mediated recruitment of Elmo. These experiments could reinforce their working hypothesis.”

We thank this reviewer for this comment. We indeed speculated too far by linking the paper from In-San Kim (*Nature Communications*, 2016), demonstrating that Stabilin-2 is a phosphatidylserine receptor for cell fusion, with the identification of a Stabilin-2/BAI3 complex in our paper. Defining if exposed “PS” would be signaling via BAI3 as part of a Stabilin-2/BAI3 complex would be a very challenge experiment to do with the current methods in place to analyze activation of BAI3. Instead, we have extensively revised our model to only focus on the data presented in our manuscript. This new model is presented in Figure 7.

“(ii) Authors suggest that “Remodeling of the cytoskeleton by the Stabilin-2/BAI3/Elmo/Dock/Rac1 pathway could be involved to drive protrusions to further increase the proximity of the myoblast membranes, similar to what is observed in *Drosophila*”. This is an important data that could improve the present work. Analysis of membrane protrusions by confocal microscopy using fluorescently labeled Phalloidin could confirm whether a similar role for actin cytoskeleton reorganization induced by the Stabilin-2/BAI3/Elmo/Dock/Rac1 pathway is observed in vertebrate muscle cells.”

We agree with the reviewer that this is an important point. However, no published work to date demonstrated a role for BAI receptors in actin nucleation. Of note, this is also true for Elmo-Dock proteins in vertebrate models. We strongly feel that this is a key question of significant importance that deserves its dedicated study in the future.

“10 – Supplementary Figure 1a,b,c – shC1qL4#2-1 and shC1qL4#2-2 do not induce a downregulation of C1qL4 mRNAs (and neither a change in cell fusion index). Authors did not discuss these results.”

Thank you for pointing to this; we also answered this for the Minor Comment #2 from Reviewer 2. We had tested 4 shRNAs. Supplementary Figure 1c-h shows that we identified a second shRNA that knocks down C1qL4 and enhances fusion (the 2 others were not efficient to knockdown C1qL4 and had no impact on fusion). We now clarified this in the manuscript and removed the irrelevant data.

“11 - Supplementary Figure 1d,e – Sol8 is a myogenic cell line isolated from primary cultures of soleus muscle with a phenotype of slow twitch fibers. Depletion of C1qL4 in these cells increases myoblast fusion. Could C1qL4 have a major role in myoblast fusion in slow fibers but not in fast twitch fibers?”

We are not aware of a report that analyzed cell fusion with respect to fiber types *in vitro*. Furthermore, there is no literature to say that satellite cells are different between fiber types. Our interpretation of the current literature is that fiber type selection occurs post-fusion and is dependent on enervation (the type of nerve that attaches to a given muscle) (see reference ¹). It is possible that C1qL4 could have a major role in a specific fiber types, but future studies would be needed to address this.

“12 – Authors did not mention in the Introduction the well-established role of cadherin/beta-catenin adhesion complexes during vertebrate myoblast fusion. They only describe the recently discovered set of vertebrate cell surface proteins that control myoblast when they mention that “DOCK1, the GTPase Rac1 and the actin nucleator N-WASP have been demonstrated to play an evolutionarily conserved and essential role in cell-cell fusion *in vivo* in mice”. Given that cadherins are highly concentrated at membrane sites of pre-fusion myoblasts, they might even interact with BAI3/Stabilin-2 membrane domains.”

This is an excellent comment and we have broadened the mention of the myoblast fusion proteins in our introduction.

REFERENCE

- 1 Feldman, J. L. & Stockdale, F. E. Skeletal muscle satellite cell diversity: satellite cells form fibers of different types in cell culture. *Dev Biol* **143**, 320-334 (1991).

Reviewers' Comments:

Reviewer #1:

Remarks to the Author:

Authors have adequately responded to all the comments. I believe manuscript is considerably improved through this revision.

Reviewer #2:

Remarks to the Author:

Overall, this is a much improved manuscript where the data support the major claims of the paper. The only piece missing from the revised MS is the in vivo relevance for C1qL4. It was asked in the first round of revision by suggesting a LOF experiment for C1qL4 in the chick embryo but it was not addressed in the rebuttal. Perhaps this experiment is not doable and should not hold up the paper given they have discussed the issue of redundancy.

I only have the following minor suggestions:

-The 'Were' beginning the last paragraph of the introduction should be 'Here'.

-The Bai3 KO data significantly adds to the paper, however there is no evidence that the reduced CSA is due to a defect in fusion. Is there a reduction of myonuclei in these mice? I think the authors should want to prove this when reporting the phenotype.

-Myomaker and Myomixer are referred to as 'pro-fusion receptors' in the second paragraph of the results section. I do not know of any data that indicates Myomaker and Myomixer are receptors.

Reviewer #3:

Remarks to the Author:

All my comments have been properly addressed and I am content with the corrections made by the authors in the new version of the manuscript. I recommend this manuscript for publication.

REVIEWER #1 and #3

“Authors have adequately responded to all the comments. I believe manuscript is considerably improved through this revision.”

“All my comments have been properly addressed and I am content with the corrections made by the authors in the new version of the manuscript. I recommend this manuscript for publication.”

We thank these reviewers for their excellent comments throughout the revision process and for the careful evaluation of our revised manuscript. We are happy that we could address all their concerns.

REVIEWER #2

“Overall, this is a much-improved manuscript where the data support the major claims of the paper. The only piece missing from the revised MS is the *in vivo* relevance for C1qL4. It was asked in the first round of revision by suggesting a LOF experiment for C1qL4 in the chick embryo but it was not addressed in the rebuttal. Perhaps this experiment is not doable and should not hold up the paper given they have discussed the issue of redundancy.”

We thank this reviewer for the constructive comment. We are sorry we have not carefully addressed the issue of LOF experiment in our rebuttal letter. Indeed, because of redundancy issue (that we had discussed in the revised version to address this point), we feel that single LOF experiments of C1qL1-4 members is unlikely to reveal clear phenotypes. Further complicating this is the fact that C1qLs are expressed broader than by the myoblasts and we may need to knockdown or knockout their expression in many tissues. In conclusion, we do not believe we will obtain interpretable data by conducting quick LOF experiments in the chick embryo model and prefer to address this in a future study. We hope the reviewer will be understanding of this decision.

“The ‘Were’ beginning the last paragraph of the introduction should be ‘Here’.”

This is now corrected. Thank you for identifying this typo.

“The Bai3 KO data significantly adds to the paper, however there is no evidence that the reduced CSA is due to a defect in fusion. Is there a reduction of myonuclei in these mice? I think the authors should want to prove this when reporting the phenotype.”

Given all the published work (the Ravichandran laboratory on BAI1 (Nature 2013) and our group on BAI3 (PNAS 2014) in addition to the data presented in this manuscript, we believe that the contribution of BAI1/3 to cell fusion is established. Nevertheless, we conducted additional experiments to address this point. We quantified the number of nuclei per fiber (revealed by Laminin staining) and these data demonstrate that there are less nuclei/fiber in Bai3-null mice. The new data is presented in Fig. 1 and Supplementary Fig. 1. We thank the reviewer for this excellent comment that improves the study.

“Myomaker and Myomixer are referred to as ‘pro-fusion receptors’ in the second paragraph of the results section. I do not know of any data that indicates Myomaker and Myomixer are receptors.”

We corrected this mistake. Thank you for pointing this out.

Reviewers' Comments:

Reviewer #2:

Remarks to the Author:

The authors have addressed the comments and this manuscript is ready for publication.

REVIEWER #2

REVIEWERS' COMMENTS:

Reviewer #2 (Remarks to the Author):

The authors have addressed the comments and this manuscript is ready for publication.

We thank the reviewer for his positive comment.